# Quantum speedup of non-linear Monte Carlo problems

**Jose Blanchet**
Stanford University
jose.blanchet@stanford.edu

**Yassine Hamoudi**
Université de Bordeaux, CNRS, LaBRI
ys.hamoudi@gmail.com

**Mario Szegedy**
Rutgers University
szegedy@cs.rutgers.edu

**Guanyang Wang**
Rutgers University
guanyang.wang@rutgers.edu

## Abstract

The mean of a random variable can be understood as a *linear* functional on the space of probability distributions. Quantum computing is known to provide a quadratic speedup over classical Monte Carlo methods for mean estimation. In this paper, we investigate whether a similar quadratic speedup is achievable for estimating *non-linear* functionals of probability distributions. We propose a *quantum-inside-quantum* algorithm that achieves this speedup for the broad class of nonlinear estimation problems known as nested expectations. Our algorithm improves upon the direct application of the quantum-accelerated multilevel Monte Carlo algorithm introduced by An et al. (2021). The existing lower bound indicates that our algorithm is optimal up to polylogarithmic factors. A key innovation of our approach is a new sequence of multilevel Monte Carlo approximations specifically designed for quantum computing, which is central to the algorithm's improved performance.

## 1 Introduction

From classic problems like Buffon's needle (Ramaley, 1969) to modern Bayesian computations (Martin et al., 2023), Monte Carlo methods have proven to be powerful tools for estimating the expected value of a given function. Specifically, the classical Monte Carlo method involves estimating $\mathbb{E}_{\mathbb{P}}[f(\mathbf{X})] = \int f(\mathbf{x}) \, d\mathbb{P}(\mathbf{x})$ by sampling, where the samples are drawn from $\mathbb{P}$. Assuming $f$ has finite variance, the number of independent and identically distributed (i.i.d.) samples required to produce an estimate of $\mathbb{E}_{\mathbb{P}}[f(\mathbf{X})]$ with an additive error of $\epsilon$ and a given degree (say 95%) of confidence is $O(1/\epsilon^2)$ [1]. In the quantum computing setting, using a Grover-type algorithm, it is known (Heinrich, 2002; Montanaro, 2015; Hamoudi and Magniez, 2019; Hamoudi, 2021; Kothari and O'Donnell, 2023) that a quantum-accelerated version of the Monte Carlo technique achieves a quadratic speedup, resulting in a cost of $O(1/\epsilon)$ for the same task. For broader links between quantum algorithms and classical stochastic simulation, see Blanchet et al. (2025).

A typical expectation $\mathbb{E}_{\mathbb{P}}[f(\mathbf{X})]$ functions as a linear functional on the space of distributions, mapping one distribution $\mathbb{P}$ to its corresponding expected value. However, many important quantities cannot be written as such linear functionals. For instance, the standard deviation maps a distribution $\mathbb{P}$ to $\mathrm{sd}(\mathbb{P}) := \sqrt{\mathbb{E}_{\mathbb{P}}[\mathbf{X}^2] - (\mathbb{E}_{\mathbb{P}}[\mathbf{X}])^2}$. This mapping is nonlinear because the expectations are subsequently transformed by quadratic and square-root operations.

---

[1] Throughout this paper, we use $f(n) = O(g(n))$ to mean $f(n) \leq M g(n)$ for all $n$ and some constant $M$. We write $f(n) = \widetilde{O}(g(n))$ to ignore polylogarithmic factors, meaning $f(n) = O(g(n) \log^k g(n))$ for some $k$.

39th Conference on Neural Information Processing Systems (NeurIPS 2025).

This work addresses settings where the target quantity is a *non-linear* functional of $\mathbb{P}$. Our goal is to estimate the *nested expectation* introduced in the next section. Our main contribution is a *quantum-inside-quantum algorithm* that achieves a near-optimal cost of $\widetilde{O}(1/\epsilon)$.

## 1.1 Nested expectation

We now establish the notation that will be used throughout the rest of the paper. Let $\mathbb{P}$ denote a probability distribution on $\mathcal{X} \times \mathcal{Y}$. Define a function $\phi : \mathcal{X} \times \mathcal{Y} \to \mathbb{R}$ and another function $g : \mathcal{X} \times \mathbb{R} \to \mathbb{R}$. Let $(X, Y) \in \mathcal{X} \times \mathcal{Y}$ be a pair of random variables with joint distribution $\mathbb{P}$. Our paper addresses the estimation of the **nested expectation**, which takes the form:

$$\mathbb{E}_X\big[g(X, \mathbb{E}_{Y|X}[\phi(X, Y)])\big]. \tag{1}$$

Equation (1) defines a non-linear function of $\mathbb{P}$, with the non-linearity arising from the non-linear nature of $g$. If we set $\gamma(x) := \mathbb{E}_{Y|X=x}[\phi(x, Y)]$ and $\lambda(x) := g(x, \gamma(x))$, the expression (1) can be written more simply as $\mathbb{E}[\lambda(X)]$. However, it is more challenging than the standard mean estimation problem as $\lambda(x)$ further depends on a conditional expectation $\gamma(x)$ that needs to be estimated. Consequently, a standard Monte Carlo method—which depends on computing $\lambda$ exactly—is not directly applicable in this context.

The phrase "nested expectation", referring to an expectation taken inside another expectation, was formally introduced and defined by Rainforth et al. (2018). It also represents the simplest nontrivial case of "repeated nested expectation" (Zhou et al., 2023; Syed and Wang, 2023; Haji-Ali and Spence, 2023). Some concrete applications are as follows:

- In Bayesian experiment design (Lindley, 1956; Hironaka and Goda, 2023), the expected information loss of the random variable $Y$ is $\mathbb{E}_X[\log(\mathbb{E}_Y[\Pr[X \mid Y]])] - \mathbb{E}_Y[\mathbb{E}_{X|Y}[\log(\Pr[X \mid Y])]]$. Here, a nested expectation appears in the first term. In typical scenarios, the conditional probability $\Pr[X \mid Y]$ has a closed-form expression, making it easy to evaluate, unlike $\Pr[X]$.

- Given $f_1, \ldots, f_d$ which can be understood as treatments, the expected value of partial perfect information (EVPPI) (Giles and Goda, 2019), is $\mathbb{E}_X[\max_k \mathbb{E}_{Y|X}[f_k(X, Y)]] - \max_k \mathbb{E}[f_k(X, Y)]$, which captures the benefit of knowing $X$. Here, a nested expectation appears also in the first term.

- In financial engineering, financial derivatives are typically evaluated using expectations and, therefore, Monte Carlo methods are often the method of choice in practice. One of the most popular derivatives is the so-called call option, whose value (in simplified form) can be evaluated as

$$\mathbb{E}_{Y|X}[\max(Y - k, 0) \mid X].$$

  Here, $k$ is the strike price, $Y$ is the price of the underlying asset at a future date, and $X$ represents the available information on the underlying. For instance, the value of a Call on a Call (CoC) option (a call in which the underlying is also a call option with the same strike price) is

$$\mathbb{E}_X[\max(\mathbb{E}_{Y|X}[\max(Y - k, 0) \mid X] - k, 0)].$$

- The objective function in conditional stochastic optimization (CSO) (Hu et al., 2020; He and Kasiviswanathan, 2024) is formulated as a nested expectation, i.e.,

$$\min_x F(x) := \min_x \mathbb{E}_\xi[f(\mathbb{E}_{\eta|\xi}[g_\eta(x, \xi)])].$$

Numerous methods are available for estimating nested expectations. The most natural way is by *nesting Monte Carlo estimators*. This method works by first sampling i.i.d. $X_1, X_2, \ldots, X_m$. For each $X_i$, one further samples $Y_{i,1}, Y_{i,2}, \ldots, Y_{i,n}$ i.i.d. from $\Pr[Y \mid X_i]$. Then $\gamma(X_i)$ can be estimated by $\widehat{\gamma}(X_i) := \sum_{j=1}^n \phi(X_i, Y_{i,j})/n$, and the final estimator is of the form

$$\frac{\sum_{i=1}^m g(X_i, \widehat{\gamma}(X_i))}{m} = \frac{\sum_{i=1}^m g(X_i, \sum_{j=1}^n \phi(X_i, Y_{i,j})/n)}{m}. \tag{2}$$

Under different smoothness assumptions on $g$, this nested estimator costs $nm = O(1/\epsilon^3)$ to $O(1/\epsilon^4)$ samples to achieve a mean square error (MSE) of up to $\epsilon^2$ (Hong and Juneja, 2009; Rainforth et al., 2018). The Multilevel Monte Carlo (MLMC) technique further improves efficiency to $O(1/\epsilon^2)$ or $O((1/\epsilon^2)\log(1/\epsilon)^2)$, as outlined in Section 9.1 of Giles (2015) and Blanchet and Glynn (2015).

If users have access to a quantum computer, An et al. (2021) proposed a QA-MLMC algorithm that improves upon the classical MLMC algorithm. The improvement ranges from quadratic to sub-quadratic—or can even be nonexistent—depending on the parameters of the MLMC framework, detailed in Sections 2.1 and 2.2.

Under the standard Lipschitz assumptions, directly applying the QA-MLMC of An et al. (2021) to our non-linear nested expectation problem incurs a cost of $\widetilde{O}(1/\epsilon^2)$. This represents no improvement over the classical cost! A technical analysis explaining the loss of the quadratic speed-up is provided in Section 3.1.

## 1.2 Our contribution

Our main algorithmic contribution of this paper is a *quantum-inside-quantum MLMC algorithm* to estimate nested expectations. Under standard technical assumptions, the algorithm achieves a cost of $\widetilde{O}(1/\epsilon)$ to produce an estimator with $\epsilon$-accuracy, which is proven to be optimal among all quantum algorithms up to logarithmic factors. As such, our algorithm provides a quadratic speedup compared to the classical MLMC algorithm, making it more efficient than the direct adaptation of the quantum algorithm proposed in An et al. (2021). The comparison between our algorithm and existing methods is summarized in Table 1.

Table 1: Cost of estimating (1) with $\epsilon$-accuracy using different methods.

| Method | Cost |
|---|---|
| Nested estimator (Eq. (2)) Rainforth et al. (2018) | $O(\epsilon^{-4})$ |
| Classical MLMC (Thm 2.1) Giles (2015) | $\widetilde{O}(\epsilon^{-2})$ |
| QA-MLMC (Thm 2.2) An et al. (2021) | $\widetilde{O}(\epsilon^{-2})$ |
| Q-NESTEXPECT (Ours) | $\widetilde{O}(\epsilon^{-1})$ |

We provide a brief explanation of the basis of our improvement and the term "quantum-inside-quantum", with more comprehensive explanations to follow in later sections. For a given target quantity and an error budget $\epsilon$, every classical MLMC algorithm consists of two components: **(1)** *decomposing* the estimation problem into the task of separately estimating the expectations of different parts; and **(2)** *distributing* the total error budget $\epsilon$ among these parts to minimize the overall computational cost, followed by using (classical) Monte Carlo to estimate each expectation. The QA-MLMC algorithm (An et al., 2021) replaces step (2) with quantum-accelerated Monte Carlo methods. However, we observed that there are multiple ways to perform the decomposition in step (1). Notably, the most natural decomposition for classical MLMC is not optimal for quantum algorithms in our nested expectation problem. To address this, we developed a new decomposition strategy based on a different sequence of quantum-accelerated Monte Carlo subroutines, ultimately achieving the desired quadratic speedup. Thus, the "outside quantum" refers to the quantum-accelerated Monte Carlo algorithm in Step (2) , while the "inside quantum" denotes the quantum subroutines used in Step (1).

Besides addressing the nested expectation problem, we show that quantum computing introduces new flexibility to MLMC. By redesigning the MLMC subroutines to leverage quantum algorithms, additional gains can be achieved. As MLMC is widely applied in computational finance, uncertainty quantification, and engineering simulations, we believe that our strategy can also improve algorithmic efficiency in many of these applications.

Finally, we stress that our quantum-accelerated Monte Carlo method uses quantum-computing algorithms to speed up the estimation of classical stochastic problems; it should not be conflated with Quantum Monte Carlo (QMC) methods, which are a separate class of classical algorithms developed for simulating quantum many-body systems (see, for example, Foulkes et al. (2001)).

The remainder of the paper is organized as follows. Section 2 reviews the fundamentals of MLMC and QA-MLMC. Section 3 outlines the limitations of existing methods, motivates our approach, and presents the new algorithm together with a proof sketch. Section 4 shows how our method accelerates several practical problems that involve nested expectations. Section 5 summarizes the work, notes its limitations, and suggests directions for future research. Detailed proofs are provided in the Appendix.

## 2 Multilevel Monte Carlo

### 2.1 Classical MLMC

MLMC methods are designed to estimate specific statistics of a target distribution, such as the expectation of a function of a Stochastic Differential Equation (SDE) (Giles, 2008), a function of an expectation (Blanchet and Glynn, 2015), or solutions to stochastic optimization problems, quantiles, and steady-state expectations (Blanchet et al., 2019). Users typically have access to a sequence of approximations, with each successive level offering reduced variance but increased computational cost. The following theorem provides theoretical guarantees for classical MLMC.

**Theorem 2.1** (Theorem 1 in Giles (2015)). *Let $\mu$ denote a quantity that we want to estimate. Suppose for each integer $l \geq 0$, we have an algorithm $\mathcal{A}_l$ that outputs a random variable $\Delta_l$ with variance $V_l$ and computational cost $C_l$. Define $s_l := \sum_{k=0}^{l} \mathbb{E}[\Delta_k]$ and assume for some $(\alpha, \beta, \gamma)$ with $\alpha \geq \frac{1}{2}\max\{\gamma, \beta\}$ that the following holds: (i) $|s_l - \mu| \leq O(2^{-\alpha l})$; (ii) $V_l \leq O(2^{-\beta l})$; (iii) $C_l \leq O(2^{\gamma l})$. Then for any fixed $\epsilon < 1/e$, one can choose positive integers $L, N_0, \ldots, N_L$ depending on $(\alpha, \beta, \gamma)$, and construct the estimator $\widehat{\mu} := \sum_{l=0}^{L} \frac{1}{N_l} \sum_{i=1}^{N_l} \Delta_l^{(i)}$ satisfying $\mathbb{E}[(\widehat{\mu} - \mu)^2] < \epsilon^2$ with cost:*

$$\sum_{l=0}^{L} C_l \cdot N_l = \begin{cases} O(\epsilon^{-2}), & \text{when } \beta > \gamma \\ O(\epsilon^{-2}(\log \epsilon)^2), & \text{when } \beta = \gamma \\ O(\epsilon^{-2-(\gamma-\beta)/\alpha}), & \text{when } \beta < \gamma, \end{cases}$$

*here $\Delta_l^{(1)}, \Delta_l^{(2)}, \ldots$ are i.i.d. copies of $\Delta_l$ for each $l$.*

The MLMC estimator described in Theorem 2.1 can be rewritten as $\widehat{\mu} = \sum_{l=0}^{L} \widehat{\delta}_l$, with each $\widehat{\delta}_l := \sum_{i=1}^{N_l} \Delta_l^{(i)} / N_l$ acting as a Monte Carlo estimator for $\mathbb{E}[\Delta_l]$.

The design and analysis of the sequence $\mathcal{A}_l$ are central to apply Theorem 2.1 in every MLMC application. In the nested expectation problem, we follow Section 9.1 of Giles (2015) and provide a possible MLMC solution here. We define $\mathcal{A}_0$ as: 1) simulate $X$; 2) simulate $Y$ given $X$; 3) output $g(X, \phi(X, Y))$. When $l \geq 1$, $\mathcal{A}_l$ is defined in Algorithm 1.

Under Assumptions 1–4 given in Section 3, one can check that the sequence of algorithms $\mathcal{A}_l$ satisfies $\alpha = 0.5$, $\beta = \gamma = 1$ in Theorem 2.1, with a proof given in Appendix B. Therefore Theorem 2.1 applies and one

---
**Algorithm 1** Classical MLMC for nested expectation: $\mathcal{A}_l$ ($l \geq 1$)

---
1: Generate $X$.
2: Generate $Y_1, \ldots, Y_{2^l}$ conditional on $X$.
3: Set $S_\text{o} = \sum_{i \text{ odd}} \phi(X, Y_i)$, $S_\text{e} = \sum_{i \text{ even}} \phi(X, Y_i)$.
4: **Output:** $g(X, 2^{-l}(S_\text{e} + S_\text{o})) - \frac{g(X, 2^{-(l-1)} S_\text{e}) + g(X, 2^{-(l-1)} S_\text{o})}{2}$.

---

can estimate the nested expectation with cost $O(\epsilon^{-2}(\log \epsilon)^2)$.

### 2.2 Quantum-accelerated MLMC in An et al. (2021)

In An et al. (2021), the authors propose a quantum-accelerated MLMC (QA-MLMC) algorithm that improves upon the classical MLMC from Theorem 2.1 in certain parameter regimes. The key insight is as follows: recall that $\widehat{\delta}_l := \sum_{i=1}^{N_l} \Delta_l^{(i)} / N_l$ is a classical Monte Carlo estimator for $\mathbb{E}[\Delta_l]$, meaning it estimates $\mathbb{E}[\Delta_l]$ through i.i.d. sampling and then takes the average. This process can be accelerated by applying quantum-accelerated Monte Carlo (QA-MC) techniques, such as those discussed in Montanaro (2015). By replacing the classical Monte Carlo estimator $\widehat{\delta}_l$ with its quantum counterpart, the following result is shown:

**Theorem 2.2** (Theorem 2 in An et al. (2021)). *With the same assumptions as in Theorem 2.1, there is a quantum algorithm that estimates $\mu$ up to additive error $\epsilon$ with probability at least $1 - \delta$, and with cost*

$$\begin{cases} \widetilde{O}(\epsilon^{-1} \log(\delta^{-1})), & \text{when } \beta \geq 2\gamma \\ \widetilde{O}(\epsilon^{-1-(\gamma-0.5\beta)/\alpha} \log(\delta^{-1})), & \text{when } \beta < 2\gamma. \end{cases}$$

Comparing QA-MLMC (Theorem 2.2) with classical MLMC (Theorem 2.1) shows only a quadratic speed-up—up to polylogarithmic factors—when $\beta \geq 2\gamma$. In a typical scenario where $\beta = \gamma = 1$

and $\alpha \geq 0.5$, classical MLMC costs $O(\epsilon^{-2})$, whereas QA-MLMC costs $\widetilde{O}(\epsilon^{-1-0.5/\alpha})$. In particular, QA-MLMC has cost $\widetilde{O}(\epsilon^{-2})$ for the nested expectation problem, using the sequence of algorithms $\mathcal{A}_l$ defined in Algorithm 1 (where $\alpha = 0.5$). The loss of any quantum speedup prompts consideration of whether the algorithm can be improved. In general, the answer is no, as we show in Appendix C that the QA-MLMC algorithm of An et al. (2021) obeys the following lower bound.

**Proposition 2.3** (Lower bound for the general QA-MLMC). *For any $(\alpha, \beta, \gamma)$, there is no quantum algorithm that can solve the problem stated in Theorem 2.1 for all sequences of algorithms $\{\mathcal{A}_l\}_{l \geq 0}$ using fewer operations than $\Omega(\epsilon^{-1})$ (when $\beta \geq 2\gamma$) or $\Omega(\epsilon^{-1-(\gamma-0.5\beta)/\alpha})$ (when $\beta \leq 2\gamma$).*

The rest of the paper is dedicated to showing that the nested expectation problem does not suffer from this lower bound, by presenting a faster quantum algorithm with cost $\widetilde{O}(\epsilon^{-1})$, but tailored to this problem.

## 3 Main Algorithm

The nested expectation $\mathbb{E}_X[g(X, \mathbb{E}_{Y|X}[\phi(X, Y)])]$ depends on $g$ and $\phi$. We define:

**Definition 3.1.** A function $g : \mathcal{X} \times \mathbb{R} \to \mathbb{R}$ is uniformly $K$-Lipschitz in the second component if there exists a positive real number $K$ such that for all $x \in \mathcal{X}$ and $y_1, y_2 \in \mathbb{R}$,

$$|g(x, y_1) - g(x, y_2)| \leq K|y_1 - y_2|.$$

We pose five assumptions:

1. The function $g$ is uniformly $K$-Lipschitz in its second component.

2. There is a number $V$ known to the users satisfying $\mathbb{E}_{Y|x}[\phi(x, Y)^2] \leq V$ for every $x \in \mathcal{X}$.

3. There is a number $S$ known to the users satisfying $\mathrm{Var}_X[g(X, \mathbb{E}_{Y|X}[\phi(X, Y)])] \leq S$.

4. We assume that users can query both $\phi$ and $g$, and that each query incurs a unit cost.

5. We have access to the following two randomized classical algorithms $\mathrm{Gen}_X$ and $\mathrm{Gen}_Y$, and every call of either one incurs a unit cost:
   (a) $\mathrm{Gen}_X$ outputs samples from the distribution of $X$ without requiring any input.
   (b) $\mathrm{Gen}_Y(x)$ accepts an input $x$ from $\mathcal{X}$ and generates a sample of $Y$ based on the conditional distribution $\Pr[Y \mid X = x]$.

We briefly review the five assumptions. The first assumption is critical for achieving the $\widetilde{O}(1/\epsilon)$ upper bound in Theorem 3.2. This assumption holds in many practical problems of interest, particularly for functions like $g(x, y) := \max\{x, y\}$ or $x + y$. The second and third assumptions are technical conditions to ensure the variance and conditional second moment are well-behaved. For example, the second assumption is satisfied when $(X, Y)$ follows a regression model $Y = f(X) + \text{Noise}$ where noise has zero mean and variance no more than $V$, and $\phi(x, y)$ is taken as $y - f(x)$. The fourth assumption is commonplace and is consistently invoked—either explicitly or implicitly—in related works (Giles and Goda, 2019; Rainforth et al., 2018; Syed and Wang, 2023).

The final assumption requires that users have complete access to the sampling procedure itself—for instance, a Python script or pseudocode that produces the distribution—rather than relying on a black-box or API that only delivers samples. This assumption is common in the simulation literature, and holds in all applications we are aware of. Under Assumption 5, if we are given a randomized algorithm that generates a random variable $X$, we can convert it into a reversible algorithm (with constant factor overhead for the time complexity) and then implement it as a quantum circuit $U : |0\rangle \mapsto \sum_x \sqrt{\Pr[X = x]}|x\rangle|\text{garbage}_x\rangle$ using classical Toffoli and NOT gates (Bennett, 1973). Here, $|\text{garbage}_x\rangle$ denotes a work tape that stores the intermediate states of the sampling procedure, ensuring that the process remains reversible.

We show:

**Theorem 3.2.** *With assumptions 1–5 stated above, for any $\delta \in (0, 0.5)$, we can design an algorithm* Q-NESTEXPECT *with cost $\widetilde{O}(K\sqrt{SV}\log(1/\delta)/\epsilon)$ that estimates $\mathbb{E}[g(X, \mathbb{E}[\phi(X, Y)])]$ with an absolute error of no more than $\epsilon$ and a success probability of $1 - \delta$.*

We highlight that the standard error metrics differ slightly between classical and quantum algorithms. Classical algorithms are often evaluated based on their expected error or mean-squared error, whereas quantum algorithms typically ensure that the error is small with high probability. The former metric is slightly stronger than the latter; however, under mild additional conditions, the two error measures become equivalent. Appendix A of An et al. (2021) has a detailed comparison between the two types of errors. Our error metric aligns with nearly all quantum algorithms, such as Brassard et al. (2002); Montanaro (2015); Kothari and O'Donnell (2023); Sidford and Zhang (2023).

## 3.1 Roadmap

**Why the quadratic speedup is lost?**

A simple yet important observation is that quantum-accelerated Monte Carlo methods typically offer a *quadratic improvement in sample complexity* compared to classical Monte Carlo methods. However, this improvement does not automatically translate to a quadratic reduction in *computational cost*. This distinction arises because the computational cost depends not only on the sample complexity but also on the cost of executing the underlying algorithm.

Suppose the goal is to estimate the expectation of a randomized algorithm $\mathcal{A}$, and the cost of a single execution of $\mathcal{A}$ is $C(\mathcal{A})$. The computational cost of classical Monte Carlo requires $O(\sigma^2/\epsilon^2)$ samples, where $\sigma^2$ represents the variance of $\mathcal{A}$, and $\epsilon$ denotes the desired error tolerance. Consequently, the total computational cost becomes $O(C(\mathcal{A})\sigma^2/\epsilon^2)$. In contrast, the QA-MC algorithm described in Kothari and O'Donnell (2023) (Theorem 3.5) reduces the sample requirement to $O(\sigma/\epsilon)$. This results in a computational cost of $\widetilde{O}(C(\mathcal{A})\sigma/\epsilon)$. This implies that QA-MC achieves a nearly quadratic speedup in computational cost *only if the cost of running the algorithm $\mathcal{A}$ is nearly constant*, i.e., $C(\mathcal{A}) = \widetilde{O}(1)$, or can itself be reduced quadratically by other methods.

The above quadratic reduction is possible when considering the standard nested Monte Carlo estimator for Equation (1), however this estimator is not competitive with the MLMC framework. It operates by estimating the inner expectation $\mathbb{E}_{Y|X=x}[\phi(X,Y)]$ with error $O(\epsilon/K)$ (which is sufficient by the Lipschitz property), nested withing an estimator of the outer expectation with error $O(\epsilon)$. The product of the sample complexity of the outer estimator $\mathcal{A}$ and the cost $C(\mathcal{A})$ of the inner estimator results in an overall complexity of $\widetilde{O}(K^2 SV/\epsilon^4)$ classically, or $\widetilde{O}(K\sqrt{SV}/\epsilon^2)$ quantumly. This is no better than the classical MLMC estimator with respect to $\epsilon$.

Now, consider the MLMC framework. Instead of using a single randomized algorithm $\mathcal{A}$, it uses a sequence of algorithms $\mathcal{A}_l$ corresponding to levels $l = 0, 1, \ldots, L$. The computational cost $C(\mathcal{A}_l)$ generally grows exponentially with $l$. For example, $C(\mathcal{A}_l) \sim 2^l$ in the MLMC solution for nested expectation problems (Algorithm 1). The highest level $L$ is chosen as $\Theta(\log(1/\epsilon))$, therefore the cost at the highest level is $C(\mathcal{A}_L) = O(\text{poly}(1/\epsilon))$.

The main idea from the QA-MLMC (An et al., 2021) is to replace each classical Monte Carlo estimator for $\mathbb{E}[\Delta_l]$ with QA-MC. While this substitution works in principle, there is a critical issue as the level $l$ approaches the maximum level $L$: the computational cost $C_l$ can grow significantly, sometimes as high as $O(\text{poly}(1/\epsilon))$. This growth undermines the quadratic speedup achieved by QA-MLMC because the total cost is no longer dominated by $O(1)$-cost subroutines. Instead, the increasing $C_l$ makes the computational cost scale poorly with the desired error tolerance $\epsilon$, leading to a loss of efficiency. Although the algorithm in An et al. (2021) leverages quantum-accelerated Monte Carlo to estimate $\mathbb{E}[\Delta_l]$, the achieved speedup is less than quadratic because $C(\mathcal{A}_l)$ scales as $O(\text{poly}(1/\epsilon))$ instead of $O(1)$ at the highest levels. This growth in cost undermines the expected quadratic speedup.

**How to recover the quadratic speedup?**

We can break down Theorem 2.1 to capture its core insights, then leverage them to refine our quantum algorithm. Theorem 2.1 provides two main ideas:

1. The target quantity is written as a limit of quantities with vanishing bias, and then re-expressed as a telescoping sum, $\mu = \lim_{l\to\infty} s_l = \sum_{l=0}^{\infty}(s_{l+1} - s_l) + s_0$. The MLMC method estimates each level individually, optimizing resource allocation.

2. Estimator Variance Control: At each level $l$, an estimator $\Delta_l$ is designed with expectation $\mathbb{E}[\Delta_l] = s_{l+1} - s_l$, and bias and variance diminishing with $l$, while cost increases with $l$.

In the classical solution (Algorithm 1), the sequence $s_l$ is

$$s_{l,\text{classical}} := \mathbb{E}\left[g\left(X, 2^{-l}\sum_{i=1}^{2^l}\phi(X, Y_i)\right)\right].$$

By the law of large numbers, it is clear that $s_{l,\text{classical}}$ converges to the target. Algorithm 1 employs standard Monte Carlo to estimate $s_{l,\text{classical}} - s_{l-1,\text{classical}}$.

A key observation is that we can design a new sequence of quantum subroutines $\mathcal{A}_{l,\text{quantum}}$. They have a computational cost similar to that of the classical $\mathcal{A}_l$ (in the sense of the $\gamma$ parameter from Theorem 2.1), while achieving improved statistical properties (parameters $\alpha$ and $\beta$ in Theorem 2.1). To achieve this goal, we first define a new sequence $s_{l,\text{quantum}}$ that also converges to the target value. Its definition is based on using quantum-accelerated Monte Carlo to estimate the *conditional expectation*, with progressively higher precision as the sequence advances. Based on this new sequence, our $\mathcal{A}_{l,\text{quantum}}$ can be naturally defined to estimate $s_{l,\text{quantum}} - s_{l-1,\text{quantum}}$. Finally, we can use existing quantum-accelerated Monte Carlo algorithms again to estimate $\mathbb{E}[\Delta_{l,\text{quantum}}]$ at each level $l$. Together, these components achieve a quadratic speedup over MLMC, making this approach provably more efficient than a direct application of QA-MLMC.

A further distinction from An et al. (2021) is the choice of the quantum subroutine: whereas An et al. (2021) builds on the QA-MC algorithm in Montanaro (2015), we instead employ the newer QA-MC algorithm of Kothari and O'Donnell (2023). The latter demands less a priori information from the user (compared to Montanaro (2015)) and removes the extra polylogarithmic factors in the error bound (compared to Hamoudi (2021)).

**Proof agenda, and the median trick**

The majority of our work is to show the following:

**Proposition 3.3.** *Under the same assumption as Theorem 2.1, we can design an algorithm* Q-NESTEXPECT-0.8 *(Algorithm 4) with cost $\widetilde{O}(K\sqrt{SV}/\epsilon)$ that estimates $\mathbb{E}[g(X, \mathbb{E}[\phi(X,Y)])]$ with an absolute error of no more than $\epsilon$ and a success probability of $0.8$.*

Proposition 3.3 specifies a fixed success probability of 0.8, whereas Theorem 3.2 allows an arbitrary value $1 - \delta$. Nevertheless, once we apply the "median trick" detailed below, we can tolerate an arbitrarily small failure probability. The proof of Lemma 3.4 is in Appendix A.4.

**Lemma 3.4** (Median trick). *Given a set of random independent random variables $X_1, \ldots, X_n$ and an unknown deterministic quantity $z$, suppose that each $X_i$ satisfies $\Pr[X_i \in (z - \delta, z + \delta)] \geq 0.8$, then $\Pr[\text{median}(X_1, \ldots, X_n) \in (z - \delta, z + \delta)] \geq 1 - \exp(-0.18n)$.*

By invoking Lemma 3.4, we can implement the Q-NESTEXPECT algorithm of Theorem 3.2 by:

Run Q-NESTEXPECT-0.8 (Algorithm 4) independently $\lceil \log(1/\delta)/0.18 \rceil$ times, and return the median of the resulting outputs.

Proving Theorem 3.2 based on Proposition 3.3 is given in Appendix A.4.

### 3.2 Our algorithm

To introduce our algorithm, we first define in Algorithm 2 a sequence of auxiliary algorithms $\mathcal{B}_l$.

---

**Algorithm 2** $\mathcal{B}_l(x)$ when $l \geq 0$

---

1: **Input:** $x$ from $X$.
2: Apply quantum-accelerated Monte Carlo Theorem 3.5 on $\text{Gen}_Y(x)$ and $\phi$ to estimate

$$\mathbb{E}_{Y|X=x}[\phi(X,Y)]$$

with accuracy $2^{-(l+1)}/K$ and success probability at least $1 - 2^{-(2l+1)}(4K^2V)^{-1}$.
3: Clip the estimate into the region $\left[-\sqrt{V}, \sqrt{V}\right]$ and denote the result by $\widehat{\phi}_l(x)$.
4: **Output:** $g(x, \widehat{\phi}_l(x))$.

---

Algorithm $\mathcal{B}_l$ with input $x$ estimates $g(x, \mathbb{E}_{Y|X=x}[\phi(X, Y)])$. Raising the value of $l$ will improve the accuracy but also elevate the computational cost. The typical way we obtain input $x$ for Algorithm $\mathcal{B}_l$ is by generating it according to $X$ using $\mathrm{Gen}_X$. We refer to this combined process as $\mathcal{B}_l(X)$. The sequence $\{\mathcal{B}_l(X)\}_{l \geq 0}$ produces random variables whose expectations progressively approximate the target $\mathbb{E}[g(X, \mathbb{E}[\phi(\bar{X}, Y)])]$. The properties of $\mathcal{B}_l$ are studied in Lemma 3.6.

Now we define $\mathcal{A}_l$ based on $\mathcal{B}_l$. When $l = 0$, we set $\mathcal{A}_0 := \mathcal{B}_0(X)$. When $l \geq 1$, we define $\mathcal{A}_l$ in Algorithm 3.

---

**Algorithm 3** $\mathcal{A}_l$ when $l \geq 1$

---

1: Sample $x$ from $X$ using $\mathrm{Gen}_X$.
2: Apply $\mathcal{B}_l(x)$ to obtain $g(x, \widehat{\phi}_l(x))$.
3: Apply $\mathcal{B}_{l-1}(x)$ to obtain $g(x, \widehat{\phi}_{l-1}(x))$.
4: **Output:** $\Delta_l := g(x, \widehat{\phi}_l(x)) - g(x, \widehat{\phi}_{l-1}(x))$.

---

Algorithm $\mathcal{A}_l$ is a "coupled" difference of $\mathcal{B}_l$ and $\mathcal{B}_{l-1}$. It executes $\mathcal{B}_l$ and $\mathcal{B}_{l-1}$ using the same input $x$. Compared to independently executing $\mathcal{B}_l$ and $\mathcal{B}_{l-1}$, this coupled implementation maintains the same expectation but ensures that the variance of $\Delta_l$ decreases to zero as $l$ increases, which is beneficial for our objective. The properties of $\mathcal{A}_l$, and of its output $\Delta_l$, are studied in Lemma 3.7.

Now we are ready to describe our solution to Proposition 3.3 in Algorithm 4.

---

**Algorithm 4** Q-NESTEXPECT-0.8

---

1: **Input:** Accuracy level $\epsilon$.
2: Set $L = \lceil \log_2(2/\epsilon) \rceil$.
3: **for** $l = 0$ to $L$ **do**
4:     Apply quantum-accelerated Monte Carlo Theorem 3.5 on $\mathcal{A}_l$ to estimate $\mathbb{E}[\Delta_l]$ with accuracy
        $\epsilon/(2L + 2)$ and success probability at least $1 - 0.1^{l+1}$. Denote the output by $\widehat{\delta}_l$.
5: **end for**
6: **Output:** $\widehat{\mu} := \sum_{l=0}^{L} \widehat{\delta}_l$.

---

Q-NESTEXPECT-0.8 is a quantum-inside-quantum algorithm as it uses quantum-accelerated Monte Carlo to estimate the expectation of another quantum-accelerated Monte Carlo algorithm. In each iteration of the for loop in Q-NESTEXPECT-0.8, we use a distinct quantum-accelerated Monte Carlo algorithm to individually estimate $\mathbb{E}[\Delta_0], \mathbb{E}[\Delta_1], \dots, \mathbb{E}[\Delta_L]$ and then sum the results.

**Proof Sketch:** To demonstrate that Algorithm 4 satisfies the guarantees stated in Proposition 3.3, we first prove a few lemmas studying the properties of $\mathcal{B}_l$ and $\mathcal{A}_l$. The crucial property we will use from Kothari and O'Donnell (2023) is as follows:

**Theorem 3.5** (Theorem 1.1 in Kothari and O'Donnell (2023)). *Given the access to an algorithm which outputs a random variable with mean $\mu$ and variance $\sigma$, there is a quantum algorithm that makes $O((\sigma/\epsilon) \log(1/\delta))$ calls of the above algorithm, and outputs an estimate $\widehat{\mu}$ of $\mu$. The estimate is $\epsilon$-close to $\mu$, that is $|\widehat{\mu} - \mu| \leq \epsilon$, with probability at least $1 - \delta$.*

Now we study the computational and statistical properties of $\mathcal{B}_l$. The formal statements of the next two lemmas can be found in Appendices A.1 and A.2.

**Lemma 3.6** (Informal). *For every $l \geq 0$, with $\mathcal{B}_l$ defined in Algorithm 2, we have:*

1. ***Cost:*** *For all $x$, the cost of implementing $\mathcal{B}_l(x)$ is at most $\widetilde{O}(2^l K \sqrt{V})$*

2. ***Mean-squared error and Bias:*** *For all $x$, the mean-squared error and the bias of the output of $\mathcal{B}_l(x)$ with respect to $g(x, \mathbb{E}_{Y|X=x}[\phi(X, Y)])$ are at most, respectively, $2^{-2l}$ and $2^{-l}$.*

3. ***Variance:*** *The variance of the output of $\mathcal{B}_l(X)$ is at most $2 + 2S$.*

Similarly, we study the properties of $\mathcal{A}_l$.

**Lemma 3.7** (Informal). *For any $l \geq 0$, with $\mathcal{A}_l$ defined in Algorithm 3, we have:*

1. **Cost:** *The cost of implementing $\mathcal{A}_l$ is at most $\widetilde{O}(2^l K \sqrt{V})$.*

2. **Bias:** *The bias of $\sum_{k=0}^{l} \Delta_k$ with respect to Equation (1) is at most $2^{-l}$.*

3. **Variance:** *The output $\Delta_l$ of $\mathcal{A}_l$ has variance at most $2 + 2S$ when $l = 0$ and $10 \times 2^{-2l}$ when $l \geq 1$.*

By applying Lemma 3.7 and ignoring lower-order terms, our new sequence $\{\mathcal{A}_l\}_{l \geq 0}$ corresponds to $\alpha = 1, \beta = 2, \gamma = 1 + o(1)$ in the MLMC framework, whereas the standard classical sequence (Algorithm 1) has $\alpha = 0.5, \beta = 1, \gamma = 1$. This improvement arises from our new quantum subroutine. It eventually yields Proposition 3.3; that result, in turn, establishes our main Theorem 3.2, demonstrating an overall complexity of $\widetilde{O}(1/\epsilon)$ when quantum-accelerated Monte Carlo is used to estimate each $\mathbb{E}[\Delta_l]$. The complete proof of Proposition 3.3 appears in Appendix A.3.

**Optimality of our algorithm:** Our algorithm achieves the optimal dependence on $\epsilon$, up to polylogarithmic factors. In particular, when $g(x, z) := z$ (which is 1-Lipschitz), the nested expectation in Equation (1) simplifies to $\mathbb{E}[\phi]$. Thus our problem includes standard Monte Carlo as a special case. Moreover, from known lower bounds on the quantum complexity of mean approximation (Nayak and Wu, 1999; Hamoudi, 2021), estimating the mean with error $\epsilon$ requires at least $\Omega(1/\epsilon)$ operations. Our algorithm matches this dependence while extending to a broader class of problems.

## 4  Applications

We now revisit the examples from Section 1.1 and analyze the cost of applying our algorithm.

**Bayesian experiment design:** Recall our target is

$$\mathbb{E}_X[\log(\mathbb{E}_Y[\Pr[X \mid Y]])] - \mathbb{E}_Y[\mathbb{E}_{X|Y}[\log(\Pr[X \mid Y])]].$$

The first term matches Equation (1) with $\phi(x, y) := \Pr[X{=}x \mid Y{=}y]$ and $g(x, z) := \log z$. Assume $\Pr[X{=}x \mid Y{=}y] \in [c, 1]$ for all $(x, y)$ (e.g., when $X$ is finite with strictly positive mass). Then Assumption 1 holds with $K = 1/c$ because the log function is $(1/c)$-Lipschitz on $[c, 1]$. Assumption 2 holds with $V = 1$ since $0 \leq \Pr[X{=}x \mid Y{=}y] \leq 1$. Assumption 3 holds with $S = \log^2(c)$ because $g(x, \cdot) = \log(\cdot) \in [\log c, 0]$. Therefore, Theorem 3.2 implies that Q-NESTEXPECT estimates the first term within $\epsilon$ error with cost $\widetilde{O}(|\log(c)|c^{-1}\log(1/\delta)/\epsilon)$.

**EVPPI:** In the EVPPI example, our target quantity is

$$\mathbb{E}_X\left[\max_{k \in \{1, \dots, d\}} \mathbb{E}_{Y|X}[f_k(X, Y)]\right] - \max_{k \in \{1, \dots, d\}} \mathbb{E}[f_k(X, Y)].$$

This problem requires a little extra care because the inner function is vector-valued in this case: $\phi(x, y) = (f_1(x, y), \dots, f_d(x, y))$ and $g(x, z) = \max(z_1, \dots, z_d)$. Ignoring the dependence on the dimension $d$, and adapting our estimator to the multidimensional case, one can estimate the first term at a cost of $\widetilde{O}(V/\epsilon)$ assuming, for instance, that the family of functions $f_k$ is uniformly bounded $V$.

**CoC option pricing:** We have $g(x, z) = \max\{z - k, 0\}$ with $\phi = g$. Then, Assumption 1 holds with $K = 1$. Assuming that both $X, Y$ are bounded between $[-a, b]$ where $a, b \geq 0$, and the strike price is $k > 0$, then Assumptions 2, 3 hold with $S = V = \max\{b - k, a + k\}^2$. Theorem 3.2 implies that Q-NESTEXPECT estimates the CoC option price $\epsilon$ error with cost $\widetilde{O}(\max\{b - k, a + k\}^2 \log(1/\delta)/\epsilon)$.

**Conditional stochastic optimization:** Recall that the goal is to find an $x$ that minimizes,

$$F(x) := \mathbb{E}_\xi[f(\mathbb{E}_{\eta|\xi}[g_\eta(x, \xi)])].$$

Assuming the decision set is finite, $\mathcal{X} = \{x_1, \dots, x_N\}$, and that each nested expectation $F(x_i)$ satisfies our Assumptions 1–5, then Q-NESTEXPECT returns an additive-$\epsilon$ estimate of $F$ at cost $\widetilde{O}(K\sqrt{SV}/\epsilon)$. This provides a function-value oracle for $F$. Using quantum minimum finding (Dürr and Høyer, 1996), the cost to output an $\hat{x}$ with $F(\hat{x}) \leq \min_i F(x_i) + \varepsilon$ is $\widetilde{O}(K\sqrt{SVN}/\epsilon)$.

# 5    Limitations and Future directions

First, although our algorithm achieves a near-optimal cost for estimating Equation (1) among quantum algorithms, our results are derived under fault-tolerance assumptions and thus remain difficult to realize on current NISQ hardware (Huang et al., 2024). Second, the original QA-MLMC algorithm applies to a wider range of scenarios than estimating the nested expectation. Thus, our contribution should be seen as an improvement of QA-MLMC within a specific yet critical set of problems.

Nevertheless, our work highlights how quantum computing can give MLMC much more freedom to invent novel approximation sequences. In favorable scenarios, like ours, a carefully crafted sequence can lead to a significantly improved (and potentially optimal) algorithm. This raises an intriguing question: which other MLMC applications could achieve a similar quantum quadratic speedup?

We expect that our technique can be extended to closely related problems. For instance, with additional assumptions, classical MLMC also has $\widetilde{O}(1/\epsilon^2)$ cost when $g(x, z) = g(z) = \mathbf{1}_{z \geq 0}$ is a Heaviside function (Giles and Haji-Ali, 2019; Gordy and Juneja, 2010). This quantity plays a key role in computing Value-at-Risk (VaR) and Conditional Value-at-Risk (CVaR), which are widely used in finance. Similarly, MLMC has been shown to be effective in conditional stochastic optimization, bilevel optimization problems (Hu et al., 2020, 2021, 2024), which is an extension of our setting with additional optimization steps. We expect our approach can be adapted to address these problems.

Our primary focus is the dependence on $\epsilon$, which is also central to both the classical (Giles, 2015) and quantum MLMC (An et al., 2021) literature. However, our problem also depends on other parameters, such as the Lipschitz constant and the variance. It can readily be extended to vector-valued random variables by using a quantum multivariate estimator (Cornelissen et al., 2022), thereby incurring a dependence on the dimension as well. An interesting open question is whether our current algorithm is optimal with respect to these parameters, or whether improvements are possible.

## Acknowledgement

J. Blanchet, M. Szegedy, and G. Wang acknowledge support from grants NSF-CCF-2403007 and NSF-CCF-2403008. Y. Hamoudi is supported by the Maison du Quantique de Nouvelle-Aquitaine "HybQuant", as part of the HQI initiative and France 2030, under the French National Research Agency award number ANR-22-PNCQ-0002.

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

# A Proofs for the Main Algorithm

## A.1 Analysis of the $\mathcal{B}_l$ algorithms

**Lemma** (Formal statement of Lemma 3.6)**.** *The algorithm $\mathcal{B}_l$ defined in Algorithm 2 for all $l \geq 0$ has the following properties:*

1. **(Cost)** *For any input $x$, the cost of implementing $\mathcal{B}_l(x)$ is $\widetilde{O}(2^l K \sqrt{V})$.*

2. **(MSE)** *For any input $x$, the mean-squared error (MSE) of the output $g(x, \widehat{\phi}_l(x))$ of $\mathcal{B}_l(x)$ with respect to $g(x, \mathbb{E}_{Y|X=x}[\phi(X,Y)])$ is at most,*

$$\mathbb{E}\left[\left|g(x, \widehat{\phi}_l(x)) - g(x, \mathbb{E}_{Y|X=x}[\phi(X,Y)])\right|^2\right] \leq 2^{-2l}.$$

3. **(Bias)** *For any input $x$, the bias of the output $g(x, \widehat{\phi}_l(x))$ of $\mathcal{B}_l(x)$ with respect to $g(x, \mathbb{E}_{Y|X=x}[\phi(X,Y)])$ is at most,*

$$\left|\mathbb{E}[g(x, \widehat{\phi}_l(x))] - g(x, \mathbb{E}_{Y|X=x}[\phi(X,Y)])\right| \leq 2^{-l}.$$

4. **(Variance)** *The variance of the output of $\mathcal{B}_l$ when the input is sampled according to $X$ is at most*

$$\mathrm{Var}_X[g(X, \widehat{\phi}_l(X))] \leq 2 + 2S.$$

*Proof of Lemma 3.6.* We prove the properties separately.

**Cost of $\mathcal{B}_l$.** Implementing Step 2–3 of Algorithm 2 incurs a computational cost of $\widetilde{O}(2^l K \sqrt{V})$. To see this, note that it would be sufficient to take $O(2^l K \sqrt{V})$ samples to have success probability 0.8 using Kothari and O'Donnell (2023). When we increase the probability of success to $1 - 2^{-(2l+1)}(4K^2V)^{-1}$, we need to multiply this with a factor of $O(\log(2^{2l+1}4K^2V))$. Thus the total cost is $\widetilde{O}(2^l K \sqrt{V})$.

**MSE:** Let $\widetilde{\phi}_l(x)$ be the output of Step 2 in $\mathcal{B}_l(x)$, i.e., the output of the quantum-accelerated Monte Carlo algorithm in Kothari and O'Donnell (2023) before clipping. We claim

$$\left|\widetilde{\phi}_l(x) - \mathbb{E}_{Y|X=x}[\phi(X,Y)]\right| \geq \left|\widehat{\phi}_l(x) - \mathbb{E}_{Y|X=x}[\phi(X,Y)]\right|,$$

i.e., clipping will never increase the error. To see this, notice that

$$\mathbb{E}_{Y|X=x}[\phi(X,Y)] \leq \sqrt{V}$$

because $\mathbb{E}[Z]^2 \leq \mathbb{E}[Z^2]$ for any random variable $Z$. Therefore, we know as a-priori that the target quantity of step 2 is between $-V$ and $V$. Suppose $\widetilde{\phi}_l(x)$ is between $[-V, V]$, then $\widetilde{\phi}_l(x) = \widehat{\phi}_l(x)$, i.e., no clipping happens. Suppose $\widetilde{\phi}_l(x) > V$, then

$$\left|\widetilde{\phi}_l(x) - \mathbb{E}_{Y|X=x}[\phi(X,Y)]\right| \geq \left|\sqrt{V} - \mathbb{E}_{Y|X=x}[\phi(X,Y)]\right| = \left|\widehat{\phi}_l(x) - \mathbb{E}_{Y|X=x}[\phi(X,Y)]\right|.$$

Similar argument holds when $\widetilde{\phi}_l(x) < -V$. Thus, the squared error either decreases or remains unchanged after clipping, establishing the claim.

Since $\widetilde{\phi}_l(x)$ has accuracy $2^{-(l+1)}/K$ with probability at least $1 - 2^{-(2l+1)}(4K^2V)^{-1}$, we know $\widehat{\phi}_l(x)$ has the same property. Moreover, the absolute error between $\widehat{\phi}_l(x)$ and $\mathbb{E}_{Y|X=x}[\phi(X,Y)]$ can be no more than $2\sqrt{V}$ due to clipping. Thus, the expected squared error between $\widehat{\phi}_l(x)$ and $\mathbb{E}_{Y|X=x}[\phi(X,Y)]$ is upper bounded by

$$\mathbb{E}\left[\left|\widehat{\phi}_l(x) - \mathbb{E}_{Y|X=x}[\phi(X,Y)]\right|^2\right] \leq K^{-2}2^{-2(l+1)} \times 1 + 4V \times 2^{-(2l+1)}(4K^2V)^{-1}$$

$$\leq K^{-2}2^{-2l}.$$

For each fixed $x$, the absolute error between $g(x, \widehat{\phi}_l(x))$ and $g(x, \mathbb{E}_{Y|x}[\phi(x, Y)])$ is not greater than

$$K|\widehat{\phi}_l(x) - \mathbb{E}_{Y|X=x}[\phi(X, Y)]|$$

due to the Lipschitzness of $g$. Therefore, we have the following estimate on the mean-squared error of $\mathcal{B}_l$ with input $x$:

$$\mathbb{E}\left[\left|g(x, \widehat{\phi}_l(x)) - g(x, \mathbb{E}_{Y|X=x}[\phi(X, Y)])\right|^2\right] \leq 2^{-2l}. \tag{3}$$

**Bias:**  The bias of the output of algorithm $\mathcal{B}_l$ on input $x$, with respect to $g(x, \mathbb{E}_{Y|X=x}[\phi(X, Y)])$, is

$$\left|\mathbb{E}[g(x, \widehat{\phi}_l(x))] - g(x, \mathbb{E}_{Y|X=x}[\phi(X, Y)])\right|$$

$$\leq \mathbb{E}\left[\left|g(x, \widehat{\phi}_l(x)) - g(x, \mathbb{E}_{Y|X=x}[\phi(X, Y)])\right|\right]$$

$$\leq \sqrt{\mathbb{E}\left[\left|g(x, \widehat{\phi}_l(x)) - g(x, \mathbb{E}_{Y|X=x}[\phi(X, Y)])\right|^2\right]}$$

$$\leq 2^{-l}.$$

Here the first two inequalities follows from $|\mathbb{E}[X] - a| \leq \mathbb{E}[|X - a|] \leq \sqrt{\mathbb{E}[|X - a|^2]}$, and the last inequality follows from Equation (3).

**Variance:**  To simplify the notation, define $G(x)$ as $g(x, \mathbb{E}_{Y|X=x}[\phi(X, Y)])$. Then Assumption 3 reads $\text{Var}_X[G(X)] \leq S$. We bound the variance of the output of $\mathcal{B}_l(X)$ as:

$$\text{Var}_X[g(X, \widehat{\phi}_l(X))] \leq \mathbb{E}_X[(g(X, \widehat{\phi}_l(X)) - \mathbb{E}_X[G(X)])^2]$$

$$\leq 2\mathbb{E}_X[(g(X, \widehat{\phi}_l(X)) - G(X))^2] + 2\mathbb{E}_X[(G(X) - \mathbb{E}_X[G(X)])^2]$$

$$\leq 2\mathbb{E}_{x \sim X}\left[\mathbb{E}[(g(x, \widehat{\phi}_l(x)) - G(x))^2]\right] + 2\text{Var}_X[(G(X)]$$

$$\leq 2 \cdot 2^{-2l} + 2S \leq 2 + 2S,$$

where the first inequality uses the fact that $\text{Var}[X] = \min_a \mathbb{E}[(X - a)^2]$ and the last inequality uses Equation (3). $\qquad\square$

## A.2  Analysis of the $\mathcal{A}_l$ algorithms

**Lemma** (Formal statement of Lemma 3.7). *The algorithm $\mathcal{A}_l$ defined in Algorithm 3 for all $l \geq 0$ has the following properties:*

1. **(Cost)** *The cost of implementing $\mathcal{A}_l$ is $\widetilde{O}(2^l K \sqrt{V})$.*

2. **(Bias)** *The bias of the sum $\sum_{k=0}^{l} \Delta_k$ of outputs up to level $l$ is at most $2^{-l}$ with respect to $\mathbb{E}_X[g(X, \mathbb{E}_{Y|X}[\phi(X, Y)])]$, i.e.:*

$$\left|\sum_{k=0}^{l} \mathbb{E}[\Delta_k] - \mathbb{E}_X[g(X, \mathbb{E}_{Y|X}[\phi(X, Y)])]\right| \leq 2^{-l}.$$

3. **(Variance)** *The variance of the output $\Delta_l$ of $\mathcal{A}_l$ is at most $2 + 2S$ when $l = 0$ and $10 \times 2^{-2l}$ when $l \geq 1$.*

*Proof of Lemma 3.7.* **Cost of $\mathcal{A}_l$.** Clearly implementing $\mathcal{A}_l$ once costs around twice of the cost of $\mathcal{B}_l$. Therefore the claims follows from Lemma 3.6.

**Bias of $\sum_{k=0}^{l} \Delta_k$.** Since the expectation of $\Delta_k$ when $k \geq 1$ is the difference in expectations between the outputs of $\mathcal{B}_k(X)$ and $\mathcal{B}_{k-1}(X)$, we know the expectation of $\sum_{k=0}^{l} \Delta_k$ is a telescoping sum that is equal to the expectation of the output of the last algorithm $\mathcal{B}_l(X)$. Therefore, the bias is

$$\left|\sum_{k=0}^{l} \mathbb{E}[\Delta_k] - \mathbb{E}[g(X, \mathbb{E}[\phi(X, Y)])]\right| = \left|\mathbb{E}_X[g(X, \widehat{\phi}_l(X)) - g(X, \mathbb{E}_{Y|X}[\phi(X, Y)])]\right|$$

$$\leq \mathbb{E}_{x \sim X}\left[\left|\mathbb{E}[g(x, \widehat{\phi}_l(x))] - g(x, \mathbb{E}_{Y|X=x}[\phi(X, Y)])\right|\right]$$

$$\leq 2^{-l}$$

where the last inequality follows from the bias established in Lemma 3.6.

**Variance of $\Delta_l$.** The case $l = 0$ follows from Lemma 3.6 and the definition $\mathcal{A}_0 = \mathcal{B}_0(X)$.

When $l \geq 1$, we can bound the second moment as follows:

$$
\begin{aligned}
\mathbb{E}[\Delta_l^2] &= \mathbb{E}_{x \sim X}\left[\mathbb{E}\left[\left(g(x, \widehat{\phi}_l(x)) - g(x, \widehat{\phi}_{l-1}(x))\right)^2\right]\right] \\
&\leq 2\mathbb{E}_{x \sim X}\left[\mathbb{E}\left[\left(g(x, \widehat{\phi}_l(x)) - g(x, \mathbb{E}_{Y|X=x}[\phi(X, Y)])\right)^2\right]\right] \\
&\quad + 2\mathbb{E}_{x \sim X}\left[\mathbb{E}\left[\left(g(x, \widehat{\phi}_{l-1}(x)) - g(x, \mathbb{E}_{Y|X=x}[\phi(X, Y)])\right)^2\right]\right] \\
&\leq 2(2^{-2l} + 2^{-2l+2}) = 10 \times 2^{-2l}
\end{aligned}
$$

where the last inequality follows from the mean-squared error established in Lemma 3.6. Since the second moment is always no less than the variance, our claim on $\mathrm{Var}[\Delta_l]$ is proven. $\qquad\square$

### A.3 Analysis of the Q-NESTEXPECT-0.8 algorithm

*Proof of Proposition 3.3.* We claim Q-NESTEXPECT-0.8 (Algorithm 4) outputs an estimator that is $\epsilon$-close to $\mathbb{E}[g(X, \mathbb{E}[\phi(X, Y)])]$ with probability at least 0.8.

For each $l$, our failure probability is no larger than $0.1^{l+1}$. Therefore the probability of at least one failure in the for-loop of Algorithm 4 is at most $0.1 + 0.1^2 + ... \leq 0.2$ by union bound. Thus there is a probability at least $1 - (0.1 + 0.1^2 + ...) \geq 0.8$ such that $\widehat{\delta}_l$ is $\epsilon/(2L + 2)$-close to $\mathbb{E}[\Delta_l]$ for every $l$. Under this (good) event, our final estimator has error

$$
\left|\sum_{l=0}^{L} \widehat{\delta}_l - \sum_{l=0}^{L} \mathbb{E}[\Delta_l]\right| \leq \sum_{l=0}^{L}|\widehat{\delta}_l - \mathbb{E}[\Delta_l]| \leq (L+1)\frac{\epsilon}{2L+2} = \frac{\epsilon}{2}.
$$

At the same time, Lemma 3.7 shows $\sum_{l=0}^{L} \mathbb{E}[\Delta_l]$ is $2^{-L}$-close to $\mathbb{E}[g(X, \mathbb{E}[\phi(X, Y)])]$. Using $L = \lceil \log_2(2/\epsilon) \rceil$, we conclude our estimator has error

$$
\begin{aligned}
\left|\sum_{l=0}^{L} \widehat{\delta}_l - \mathbb{E}[g(X, \mathbb{E}[\phi(X, Y)])]\right| &\leq \left|\sum_{l=0}^{L} \widehat{\delta}_l - \sum_{l=0}^{L} \mathbb{E}[\Delta_l]\right| \\
&\quad + \left|\sum_{l=0}^{L} \mathbb{E}[\Delta_l] - \mathbb{E}[g(X, \mathbb{E}[\phi(X, Y)])]\right| \leq \epsilon,
\end{aligned}
$$

with probability at least 0.8.

The cost of our algorithm is the summation of the costs of $L + 1$ different quantum-accelerated Monte Carlo algorithms. For $l = 0$, since $\mathcal{A}_0 = \mathcal{B}_0(X)$, the variance of $\Delta_0$ is at most $2 + 2S$ by Lemma 3.6. Thus, Theorem 3.5 shows that the cost of estimating $\mathbb{E}[\Delta_0]$ with error $\epsilon/(2L + 2)$ is $\widetilde{O}(\sqrt{S}L/\epsilon)$ times the cost $\widetilde{O}(K\sqrt{V})$ of implementing $\mathcal{B}_0(X)$, which gives $\widetilde{O}(LK\sqrt{SV}/\epsilon)$. For $l \geq 1$, the variance of $\Delta_l$ is at most $10 \times 2^{-2l}$ by Lemma 3.7. Hence, the cost of estimating $\mathbb{E}[\Delta_l]$ is $\widetilde{O}(2^{-l}L/\epsilon)$ times the cost $\widetilde{O}(2^l K\sqrt{V})$ of implementing $\mathcal{A}_l$, which gives $\widetilde{O}(LK\sqrt{V}/\epsilon)$.

Summing up all the costs together, and using that $L = O(\log(1/\epsilon))$, yields a total complexity of

$$
\widetilde{O}(LK\sqrt{SV}/\epsilon + L^2 K\sqrt{V}/\epsilon) = \widetilde{O}(K\sqrt{SV}/\epsilon).
$$

$\qquad\square$

### A.4 Median trick and main theorem

*Proof of Lemma 3.4 (Median trick).* Let

$$
Y_i = \begin{cases} 1, & \text{if } X_i \notin (z - \delta, z + \delta), \\ 0, & \text{otherwise,} \end{cases} \qquad i = 1, \ldots, n,
$$

and set $S_n = \sum_{i=1}^n Y_i$. Because each $X_i$ lies in the interval $(z - \delta, z + \delta)$ with probability at least 0.8, we have $\Pr[Y_i = 1] \le 0.2$ and hence

$$\mathbb{E}[S_n] = \sum_{i=1}^n \mathbb{E}[Y_i] \le 0.2n.$$

The sample median falls *outside* the interval $(z - \delta, z + \delta)$ if and only if more than half of the observations do so, i.e., if $S_n > \frac{1}{2}n$. Consequently,

$$\Pr[\mathrm{median}(X_1, \ldots, X_n) \notin (z - \delta, z + \delta)] = \Pr\left[S_n > \frac{1}{2}n\right] \le \Pr[S_n - \mathbb{E}[S_n] \ge 0.3n].$$

Because the $Y_i$ are independent Bernoulli variables, Hoeffding's inequality gives

$$\Pr[S_n - \mathbb{E}[S_n] \ge t] \le \exp(-2t^2/n), \qquad t > 0.$$

Taking $t = 0.3n$ yields

$$\Pr[S_n > \frac{1}{2}n] \le \exp(-2(0.3n)^2/n) = \exp(-0.18n).$$

Therefore,

$$\begin{aligned}
\Pr[\mathrm{median}(X_1, \ldots, X_n) \in (z - \delta, z + \delta)] &= 1 - \Pr[\mathrm{median}(X_1, \ldots, X_n) \notin (z - \delta, z + \delta)] \\
&\ge 1 - \exp(-0.18n),
\end{aligned}$$

as claimed. $\qquad\square$

The proof of Theorem 3.2 follows immediately by applying the median trick to the algorithm given in Proposition 3.3.

*Proof of Theorem 3.2.* The Q-NESTEXPECT algorithm was defined as,

> Run Q-NESTEXPECT-0.8 (Algorithm 4) independently $\lceil \log(1/\delta)/0.18 \rceil$ times, and return the median of the resulting outputs.

Since the output of Q-NESTEXPECT-0.8 is $\epsilon$-close to Equation (1) with probability at least 0.8 (Proposition 3.3), the median trick (Lemma 3.4) implies that the output of Q-NESTEXPECT is $\epsilon$-close to Equation (1) with probability at least $1 - \exp(-0.18\lceil \log(1/\delta)/0.18 \rceil) \ge 1 - \delta$. The cost of Q-NESTEXPECT is also $\lceil \log(1/\delta)/0.18 \rceil$ times the cost of Q-NESTEXPECT-0.8, and is thus $\widetilde{O}(K\sqrt{SV}\log(1/\delta)/\epsilon)$, as claimed. $\qquad\square$

## B  Classical MLMC Parameters

Under Assumptions 1–4 in Section 3, we verify that the classical MLMC sequence defined in Algorithm 1 corresponds to $\alpha = 0.5, \beta = \gamma = 1$ in the MLMC framework (Theorem 2.1) when $K, V = O(1)$ are constant numbers. Firstly, the computational cost of $\mathcal{A}_l$ is clearly $O(2^l)$, thus $\gamma = 1$. To analyze the variance, it is useful to observe

$$\left| g\left(X, \frac{1}{2^l}(S_e + S_o)\right) - \frac{g\left(X, \frac{1}{2^{l-1}}S_e\right) + g\left(X, \frac{1}{2^{l-1}}S_o\right)}{2} \right| \le \frac{K}{2^{l-1}}|S_e - S_o|$$

because of the Lipschitz continuity. Let $Z_i = \phi(X, Y_{2i-1}) - \phi(X, Y_{2i})$, it is then clear that the $Z_i$'s have zero mean and are conditionally independent given $X$. Further,

$$\mathrm{Var}[Z_i \mid X] = \mathrm{Var}[\phi(X, Y_{2i-1}) \mid X] + \mathrm{Var}[\phi(X, Y_{2i}) \mid X] = 2\mathrm{Var}_Y[\phi(X, Y) \mid X] \le 2V.$$

Therefore

$$\mathbb{E}[|S_e - S_o|^2] = \mathrm{Var}[S_e - S_o] = \mathrm{Var}\left[\sum_{i=1}^{2^{l-1}} Z_i\right] \le 2^l V.$$

Thus

$$\mathbb{E}\left[\left|g\left(X, \frac{1}{2^l}(S_{\mathrm{e}} + S_{\mathrm{o}})\right) - \frac{g\left(X, \frac{1}{2^{l-1}}S_{\mathrm{e}}\right) + g\left(X, \frac{1}{2^{l-1}}S_{\mathrm{o}}\right)}{2}\right|^2\right] \leq \frac{K^2}{2^{2l-2}}\mathbb{E}[|S_e - S_o|^2]$$

$$\leq \frac{K^2}{2^{2l-1}}2^l V$$

$$= \frac{2K^2 V}{2^l}$$

$$= O(2^{-l}).$$

Therefore $\beta = 1$, as claimed.

Finally we study the bias, which is

$$b := \left|\sum_{k=0}^{l} \mathbb{E}[\Delta_k] - \mathbb{E}[g(X, \mathbb{E}[\phi(X, Y)])]\right|.$$

where $\Delta_0 = g(X, \phi(X, Y))$ and $\Delta_k = g(X, 2^{-k}(S_{\mathrm{e}} + S_{\mathrm{o}})) - \frac{g(X, 2^{-(k-1)}S_{\mathrm{e}}) + g(X, 2^{-(k-1)}S_{\mathrm{o}})}{2}$ when $k \geq 1$. We obtain a telescoping sum simplifying to

$$\sum_{k=0}^{l} \mathbb{E}[\Delta_k] = \mathbb{E}_X\left[g\left(X, \frac{1}{2^l}(S_{\mathrm{e}} + S_{\mathrm{o}})\right)\right] = \mathbb{E}_{X, Y_1, \ldots, Y_{2^l}}\left[g\left(X, \frac{\sum_{k=1}^{2^l} \phi(X, Y_k)}{2^l}\right)\right].$$

Therefore, the bias is at most

$$b \leq \mathbb{E}_{X, Y_1, \ldots, Y_{2^l}}\left[\left|g\left(X, \frac{\sum_{k=1}^{2^l} \phi(X, Y_k)}{2^l}\right) - g(X, \mathbb{E}_{Y|X}[\phi(X, Y)])\right|\right]$$

(by the triangle inequality)

$$\leq K\mathbb{E}_{X, Y_1, \ldots, Y_{2^l}}\left[\left|\frac{\sum_{k=1}^{2^l} \phi(X, Y_k)}{2^l} - \mathbb{E}_{Y|X}[\phi(X, Y)]\right|\right] \qquad \text{(since } g \text{ is } K\text{-Lipschitz)}$$

$$\leq K\mathbb{E}_X\left[\sqrt{\mathbb{E}_{Y_1, \ldots, Y_{2^l}|X}\left[\left|\frac{\sum_{k=1}^{2^l} \phi(X, Y_k)}{2^l} - \mathbb{E}_{Y|X}[\phi(X, Y)]\right|^2\right]}\right]$$

(by the Cauchy-Schwarz inequality)

$$= \frac{K}{2^{l/2}}\mathbb{E}_X\left[\sqrt{\mathrm{Var}_{Y|X}[\phi(X, Y)]}\right]$$

$$\leq \frac{K\sqrt{V}}{2^{l/2}} \qquad \text{(by Assumption 2 in Section 3)}$$

Thus, we have $\alpha = 0.5$, as claimed.

## C  Lower Bound for the QA-MLMC Algorithm of An et al. (2021)

We demonstrate Proposition 2.3, namely that the QA-MLMC algorithm of An et al. (2021) is optimal (up to polylogarithmic factors) for solving the problem stated in Theorem 2.1 when the sequence $\{\mathcal{A}_l\}_{l \geq 0}$ is arbitrary. We consider two parameter regimes for $(\alpha, \beta, \gamma)$ separately.

**Regime $\beta \leq 2\gamma$.**  In that regime, the complexity of the quantum MLMC algorithm of An et al. (2021) is $\widetilde{O}(\epsilon^{-1-(\gamma-0.5\beta)/\alpha})$. We observe that the most expensive part of the computation in this algorithm occurs at the last level $L = O(\log(1/\epsilon)/\alpha)$, where the algorithm uses $\widetilde{O}(\epsilon^{-1+0.5\beta/\alpha})$ samples, each with cost $C_L = \widetilde{O}(\epsilon^{-\gamma/\alpha})$. We take inspiration from this behavior to design a hard instance for the lower bound. The lower bound proceeds by a reduction to the following query problem:

**Definition C.1** ($\text{COUNT}^{t,n} \circ \text{PARITY}^m$). Fix three integers $t, n, m$ with $t \leq n$. Define two sets of Boolean matrices $\mathbf{M}_0, \mathbf{M}_1 \subset \{0,1\}^{n \times m}$ such that:

- $M \in \mathbf{M}_0$ iff each row of $M$ has parity 0 (i.e., $\sum_j M_{i,j} = 0 \mod 2$ for all $i \in [n]$),

- $M \in \mathbf{M}_1$ iff exactly $t$ rows of $M$ have parity 1.

In the $\text{COUNT}^{t,n} \circ \text{PARITY}^m$ problem, one is given access to a quantum oracle $|i,j\rangle \mapsto (-1)^{M_{i,j}}|i,j\rangle$ for an unknown $M \in \mathbf{M}_0 \cup \mathbf{M}_1$, and the goal is to decide to which set $M$ belongs to.

By composition properties of the quantum adversary method, the quantum query complexity of $\text{COUNT}^{t,n} \circ \text{PARITY}^m$ is the product of the complexity of the counting problem $\text{COUNT}^{t,n}$ (which is $\Omega(\sqrt{n/t})$) and of the parity problem $\text{PARITY}^m$ (which is $\Omega(m)$). We refer the reader to (van Apeldoorn, 2021, Lemma 11) for a more detailed version of this argument.

**Proposition C.2.** *Any quantum algorithm that solves the* $\text{COUNT}^{t,n} \circ \text{PARITY}^m$ *problem with success probability at least* $2/3$ *must make* $\Omega(m\sqrt{n/t})$ *queries to* $M$.

We now explain how to construct a hard instance for the MLMC problem out of an input to the $\text{COUNT}^{t,n} \circ \text{PARITY}^m$ problem. Fix $(\alpha, \beta, \gamma)$ such that $\alpha \geq \frac{1}{2}\max\{\gamma, \beta\}$ and $\beta \leq 2\gamma$. Choose $t, n, m$ such that $m = C_L = \epsilon^{-\gamma/\alpha}$ and $t/n = \epsilon^2/V_L = \epsilon^{2-\beta/\alpha}$. Given $M \in \mathbf{M}_0 \cup \mathbf{M}_1$, define the random variable $P_M \in \{0, \epsilon^{\beta/\alpha-1}\}$ obtained by sampling $i \in [n]$ uniformly at random and setting

$$P_M = \begin{cases} 0, & \text{if } \text{PARITY}(M_i) = 0, \\ \epsilon^{\beta/\alpha-1}, & \text{if } \text{PARITY}(M_i) = 1. \end{cases}$$

We consider the problem of estimating the expectation $\mu = \mathbb{E}[P_M]$ with error $\epsilon/2$. Observe that $\mu = 0$ if $M \in \mathbf{M}_0$ and $\mu = t/n\epsilon^{\beta/\alpha-1} = \epsilon$ if $M \in \mathbf{M}_1$. Hence, estimating $\mu \pm \epsilon/2$ allows to solve the $\text{COUNT}^{t,n} \circ \text{PARITY}^m$ problem on $M$. By Proposition C.2, this cannot be done faster than with $\Omega(m\sqrt{n/t}) = \Omega(\epsilon^{-1-(\gamma-0.5\beta)/\alpha})$ quantum queries.

It remains to construct a sequence of random variables $\Delta_l$ with parameters $(\alpha, \beta, \gamma)$ that can approximate $P_M$. All random variables are chosen to be zero, except $\Delta_L$ which is chosen to be $P_M$ (where $L = \log(1/\epsilon)/\alpha$). The cost to compute $\Delta_L$ (naively) is $C_L = m = \epsilon^{-\gamma/\alpha} = 2^{\gamma L}$. The variance is $\text{Var}[\Delta_L] = \text{Var}[P_M] \leq \mathbb{E}[P_M^2] \leq \epsilon^{\beta/\alpha} = 2^{-\beta L}$. The cost and variance of $\Delta_l$ are zero when $l \neq L$. Finally, for any $l$ we have,

$$\left| \sum_{k=0}^{l} \mathbb{E}[\Delta_k] - \mu \right| = \begin{cases} 0, & \text{if } l \geq L, \\ \mu \leq \epsilon \leq 2^{-\alpha l}, & \text{if } l < L. \end{cases}$$

Hence, we have constructed a sequence of algorithms $\{\mathcal{A}_l\}_{l \geq 0}$ satisfying the conditions listed in Theorem 2.1, but for which no quantum algorithm can solve the estimation problem using fewer than $\Omega(\epsilon^{-1-(\gamma-0.5\beta)/\alpha})$ quantum queries.

**Regime** $\beta \geq 2\gamma$. In that regime, the complexity of the quantum MLMC algorithm of An et al. (2021) is $\widetilde{O}(\epsilon^{-1})$. The most expensive part of the computation occurs at the first level $l = 0$, where the algorithm uses $\widetilde{O}(\epsilon^{-1})$ samples, each with cost $C_L = \widetilde{O}(1)$. This is trivial to adapt into a lower bound (in fact, it applies to the regime $\beta \leq 2\gamma$ as well). We know that for standard Monte Carlo, there exists a quantum query problem (Nayak and Wu, 1999; Hamoudi, 2021) for which generating a certain random variable $X$ has cost $O(1)$ and variance $\text{Var}[X] = O(1)$, and the quantum complexity of estimating $\mathbb{E}[X] \pm \epsilon$ is $\Omega(1/\epsilon)$. This can be cast as an MLMC problem by considering only the first level: $\Delta_0 = X$. We have $\left| \sum_{k=0}^{l} \mathbb{E}[\Delta_k] - \mathbb{E}[X] \right| = 0$ for all $l$, and $C_l = V_l = 0$ for all $l > 0$. Since this sequence applies to any values of $(\alpha, \beta, \gamma)$, we have a generic lower bound of $\Omega(1/\epsilon)$.

