# OpenReview forum: "Quantum speedup of non-linear Monte Carlo problems"
_NeurIPS.cc/2025/Conference — NeurIPS 2025 spotlight_

### Official Review · Reviewer_BADi · 2025-06-22

**Clarity:** 3
**Significance:** 2
**Originality:** 2
**Rating:** 4
**Confidence:** 4

**Summary:**

This paper studies the nested expectation estimation problem and proposes a new quantum algorithm that achieves a quadratic quantum speedup over the classical multi-level Monte Carlo method. It also improves the previous QA-MLMC by An et al.

**Questions:**

Why does the equation below Line 544 hold?

Which assumptions in Lines 166-175 are possible to relax?

**Ethical Concerns:**

["NO or VERY MINOR ethics concerns only"]

**Final Justification:**

I agree with the authors that this paper presents a new "quantum-quantum" structure for MLMC, which could be useful in certain specific research areas. However, the technical contributions remain weak, and the algorithm might have limited practical impact. Therefore, I believe this paper is on the borderline of acceptance.

**Limitations:**

yes

**Quality:**

3

**Strengths And Weaknesses:**

Strengths: This paper is well-written and all the theorems/lemmas are mathematically sound to me. The new quantum algorithm enhances the previous QA-MLMC in two aspects: 1. It utilizes a state-of-the-art quantum mean estimation procedure that has several beneficial properties compared to the version used by An et al. 2. In the conditional expectation estimation phase, directly applying An et al. employs a classical procedure, while the new quantum algorithm integrates the quantum mean estimation.This reduces the complexity from $\epsilon^{-1.5}$ to $\epsilon^{-1}$, which is optimal in quantum.

Weaknesses: First, the idea of the new quantum algorithm is quite natural and straightforward (although the analysis contains some non-trivial work). More specifically, the improvement over QA-MLMC highly depends on the structure of the problem studied in this paper. Therefore, the technical contribution of this paper is relatively weak. Second, the previous work, QA-MLMC, has wider applications than the algorithm in this paper. To apply the algorithm proposed in this paper, five assumptions are required; however, these may not hold in certain practical scenarios. To strengthen this paper, it would be better to provide more detailed discussions regarding the applications in  Bayesian experiment design, EVPPI, and CoC. More specifically, it may provide more explicit calculations regarding the costs and speedups by taking into account certain practical parameters of these problems.

---

> ### Author Rebuttal · Authors · 2025-07-28
>
> We thank the reviewer for the careful reading! We are encouraged that the reviewer thinks the paper is well-written and mathematically sound. We address the questions below:
>
> **On Technical Contribution and Originality**
>
> We appreciate and agree the reviewer's perspective that the high-level idea is natural.
>
> We would like to clarify that our primary technical contribution is not simply the substitution of a procedure with a quantum counterpart, but the development of a new "quantum-inside-quantum" structure that is necessary to achieve the optimal speedup.
>
> From a complexity-theoretic standpoint, our work helps characterize the quantum query complexity of this problem by providing a near-optimal algorithm, which provides new insights to understand this problem.
>
> From the method perspective, the previous QA-MLMC algorithm applies quantum speedup only at the "outer" level. A direct application of this method to our problem results in a suboptimal cost of $\tilde O(\epsilon^{−1.5})$ because the cost of the classical inner-level estimators grows too quickly. Our key insight is that one must redesign the MLMC levels themselves by creating a new sequence of quantum subroutines that leverage quantum estimation for the inner conditional expectation. We believe this demonstrates a novel design pattern for quantum MLMC that is a valuable methodological improvement beyond prior work.
>
> **Comparison with QA-MLMC**
> The reviewer correctly notes that our algorithm is more specialized than the general QA-MLMC framework. This was a deliberate trade-off. Our goal was to demonstrate that for the important class of nested expectation problems, a full quadratic speedup is achievable. We have traded generality for optimality in this critical domain.
>
> **Assumptions**
>
> Great point. Among the five assumptions, assumption 4-5 are essential to setup the problem, and is explicitly or implicitly assumed in the relevant literature that we are aware of.
>
> Regarding assumption 1-3, we acknowledge that they may not hold perfectly or may be hard to check in some practical scenarios. Nevertheless, they are also standard assumptions in both the classical and quantum literature, for example, the smoothness assumption is made in [1-6]. That said, we agree with the reviewer that these assumptions may not hold in certain applications.
>
> Following the reviewer's suggestion, we will expand Section 4 in our revision. *We will provide more explicit cost calculations for applications like EVPPI*, detailing how practical parameters influence the overall speedup and making the significance of our result more concrete. We also discuss possible ways to relax these assumptions (see the response to the last question), and will comment on them in the revised version.
>
> **Derivation of line 544**:
> We first note that $\mathbb E_{Y\mid X}[S_{\text{E}}] = \mathbb E_{Y\mid X}[S_{\text{O}}] = 2^{l-1}\mathbb E_{Y\mid X}[\phi(X,Y)]$ (by our definition in Algorithm 1). Therefore,
>
> $$\mathbb E_{Y\mid X}\left[\left\lVert\frac{1}{2^l} (S_{\text{E}} + S_{\text{O}})  - \mathbb E_{Y\mid X}[\phi(X,Y)] \right\rVert\right] \leq \mathbb E_{Y\mid X}\left[\left\lVert\frac{1}{2^l} (S_{\text{E}} + S_{\text{O}})  -  \mathbb E_{Y\mid X}[\phi(X,Y)]\right\rVert^2\right]^{0.5} $$
>
> by Cauchy-Schwarz. The right-hand side further simplifies to $\lVert Var_{Y\mid X}\left[\frac{1}{2^l}(S_{\text{E}} + S_{\text{O}})  -\mathbb E_{Y\mid X}[\phi(X,Y)]\right]\rVert_1$ (since the random vector inside has an expectation of 0 for each coordinate, so its second moment equals the variance). Here, variance of a random vector is taken on each coordinate, just as we did for expectation. Note that there is a typo in the right-hand side of the first inequality under line 544, where there is an additional $\lVert \cdot \rVert$ in the variance. It should be $\lVert \cdot \rVert_1$ outside the variance. We'll fix this typo in the revision.
>
> Since given $X$, $\mathbb E_{Y\mid X}\left[\phi(X,Y)]\right]$ is a constant that does not change the variance, we can further simply the variance term to $Var_{Y\mid X}\left[\frac{1}{2^l}(S_{\text{E}} + S_{\text{O}})]\right]$.
>
> We further claim $$ Var_{Y\mid X}\left[\frac{1}{2^l}(S_{\text{E}} + S_{\text{O}})]\right]  = \frac 1 {2^l} Var_{Y\mid X}(\phi(X, Y)),$$
> because both $S_{\text{E}}$ and $S_{\text{O}}$ are sum of $2^{l-1}$ i.i.d. random vectors each has the same distribution as $\phi(X,Y)$, therefore averaging over $2^l$ i.i.d. copy reduce the variance of each copy by a factor of $2^{-l}$. Plugging into the earlier inequality,
>
> $$\mathbb E_{Y\mid X}\left[\left\lVert\frac{1}{2^l} (S_{\text{E}} + S_{\text{O}})  - \mathbb E_{Y\mid X}[\phi(X,Y)]] \right\rVert\right]  \leq \frac{1}{2^{l/2}} \lVert Var_{Y\mid X}(\phi(X,Y))\rVert_1^{0.5} \leq \frac{m^{0.5} V^{0.5}}{2^{l/2}}$$ by Assumption 2.
>
>
> **Which assumptions are possible to relax**:
>
> The question is also asked by reviewer hqsW. There are essentially two types of technical assumptions appeared in the current version:
> 1) *Smoothness assumption (Assumption 1);*
> 2) *Moment assumption (Assumption 2,3).*
>
> We discuss them separately.
>
> *Regarding the smoothness assumption*: It is indeed possible to generalize this assumption. Additional knowledge about the functions or the underlying distribution could further relax the continuity requirement. In classical Monte Carlo literature, there are studies on nested expectations where the function $g$ in our notation is a Heaviside function [7,8]. The idea of their assumptions is to assume that the underlying distribution behaves appropriately at the discontinuous points, ensuring that it does not cause issues.
> In summary, there is a tradeoff between the assumptions on distributions and the regularity assumptions on the functions. Stronger distributional assumptions and/or extra knowledge on the function can lead to weaker assumptions on its smoothness, and vice versa. If we use the same distribution assumptions as in [3,4] and assume $g$ is a Heaviside function, we could still get the quadratic speedup.
>
> *On the necessity of finite second moments:* A finite second moment may be required to achieve a complexity of $\tilde{O}(\epsilon^{-1})$. Reference [9] shows, even in the context of standard Monte Carlo estimation, assuming only the existence of a $(1 + \delta)$-th moment with $\delta \in (0,1)$ is insufficient for attaining $\tilde{O}(\epsilon^{-1})$ cost with a quantum algorithm (but a quadratic speedup still exists). Therefore one may still hope for a quadratic speedup for the nested expectation problem without finite second moment, however, the quantum algorithm's cost will also be worst than $\tilde{O}(\epsilon^{-1})$.
>
> *On the known constants for the second moment/variance:* There exists an interesting intermediate regime where the assumption (1-2) could be replaced by $<\infty$ (instead of the conditions $\leq V$ and $\leq S$, where $V$ and $S$ are constants known to the users in advance.) Right now we do not know if we can design a quantum algorithm to achieve $\tilde O(\epsilon^{-1})$ cost in this relaxed assumption.
> One potential approach is to first perform a low-cost Monte Carlo estimation to obtain an estimated upper bound that holds with high probability (e.g., via the Markov inequality), and then plug-in this estimated bound into our algorithm.
> A rigorous analysis of these costs is likely complex, but it is certainly a worthwhile direction for future work. In particular, we note that the assumption of finite variance with a known constant is also common in other quantum Monte Carlo applications, such as stochastic optimization [4]. Therefore, relaxing these assumptions could be beneficial, potentially strengthening the results in these contexts as well.
>
> We hope these responses clarifies your concerns and questions. Thanks again for your valuable feedback!
>
>
>
>
> [1] On nesting Monte Carlo estimators, Rainforth et. al., ICML 2018.
>
> [2] Multilevel Simulation of Functionals of Bernoulli Random Variables with Application to Basket Credit Derivatives, Bujok et. al., Methodology and Computing in Applied Probability, 2013.
>
> [3] Optimal randomized multilevel Monte Carlo for repeatedly nested expectations, Syed and Wang, ICML 2023
>
> [4] Quantum speedups for stochastic optimization, Sidford and Zhang, NeurIPS 2023.
>
> [5] Biased Stochastic First-Order Methods for Conditional Stochastic Optimization and Applications in Meta Learning, Hu et. al., NeurIPS 2020.
>
> [6] Bias-Variance-Cost Trade-off of Stochastic Optimization, Hu et. al., NeurIPS 2021.
>
> [7] Multilevel Nested Simulation for Efficient Risk Estimation, Giles and Haji-Ali, SIAM Journal on Uncertainty Quantification 2019.
>
> [8] Nested Simulation in Portfolio Risk Measurement, Gordy and Juneja, Management Science, 2010
>
> [9] Quadratic Speed-up in Infinite Variance Quantum Monte Carlo, Blanchet et. al. 2024.

---

> > ### Comment · Reviewer_BADi · 2025-08-04
> >
> > We appreciate the authors' detailed response. Most of my concerns have been resolved or clarified. I'll raise my score.

---

### Official Review · Reviewer_QYbp · 2025-07-01

**Clarity:** 4
**Significance:** 4
**Originality:** 3
**Rating:** 5
**Confidence:** 4

**Summary:**

The paper provides a quantum algorithm for the Monte-Carlo estimation of nested quantities. This nested MC considers the evaluation of an expectation value of a function that depends on another expectation value.

**Questions:**

Is it possible to reason about the implications for end-to-end advantages?

**Ethical Concerns:**

["NO or VERY MINOR ethics concerns only"]

**Final Justification:**

It is a solid work that advances the research on where amplitude amplification/estimation could lead to polynomial quantum advantages.

**Limitations:**

Yes.

**Paper Formatting Concerns:**

No.

**Quality:**

4

**Strengths And Weaknesses:**

Strengths: The quantum algorithm achieves the near-optimal result 1/eps via a clever distribution of the error to the parts of the Monte Carlo estimation and the quantum subroutines. The paper contains a mathematical solid discussion of the algorithm and its guarantees. The paper is an important result about the various ways to use amplitude amplification and estimation towards practical applications. It has implications to several domains such as finance, AI, and optimization.

Weaknesses: The paper is fully theoretical and does not contain numerics. However, the fundamental nature of the work and the results justify this presentation. The paper relies on the inputs given via oracles, which is standard for similar works, but leaves end-to-end advantages for future work.

---

> ### Author Rebuttal · Authors · 2025-07-28
>
> We thank the reviewer for their positive assessment and insightful comments. We are delighted the reviewer recognized the technical strength and potential impact of our work.
>
> The reviewer is correct in noting that our work uses the standard oracle model. A full end-to-end speedup, which includes the oracle construction, should be theoretically feasible. Theoretically, an end-to-end speedup depends on efficient state preparation. Assuming we have access to a classical algorithm that prepares the underlying distribution of \$X\$ and \$Y \mid X\$ (which is practical in the applications we considered), we can convert it into a quantum oracle using classical Toffoli and NOT gates with constant overhead [1]. In practice, it will involve additional complexities, particularly around error correction.
>
> We will include more discussions on end-to-end advantages in the revised version. Thanks again for the valuable feedback, we remain available for further questions.
>
> [1] Logical reversibility of computation, Bennett, IBM journal of Research and Development, 1973.

---

### Official Review · Reviewer_H1SM · 2025-07-01

**Clarity:** 3
**Significance:** 3
**Originality:** 3
**Rating:** 5
**Confidence:** 3

**Summary:**

This paper proposes a quantum algorithm for non-linear nested expectation values The approach builds upon quantum-accelerated Monte Carlo, which provides a quadratic speedup for multi-level Monte Carlo (MLMC) methods. The main innovation is the use of quantum-inside-quantum, which allows their algorithm to retain the quadratic speedup in contrast to a naive application of quantum-accelerated MLMC. It is shown that the algorithm achieves near-optimal complexity of $\widetilde{O}(1/\epsilon)$. Several applications of nested expectations are briefly mentioned.

**Questions:**

- While the algorithm is optimal due to the worst-case lower bound for mean approximation, are there any applications where the best known classical algorithm has cost strictly greater than $\mathcal{O}(1/\epsilon^2)$, i.e., the quantum algorithm achieves more than a quadratic speedup in practice?
- The main innovation of this paper is that both the outer and inner loops of classical MLMC for nested expectations must be accelerated to achieve the full quadratic speedup. However, intuitively it seems like nested loops should be able to be unraveled into a single loop, in the same way that a quantum-inside-quantum algorithm can be viewed as a single quantum algorithm. Is there a way to re-interpret Q-NestExpect as a singly quantum-accelerated Monte Carlo algorithm?

**Ethical Concerns:**

["NO or VERY MINOR ethics concerns only"]

**Final Justification:**

I think this paper has a result that is both theoretically interesting and potentially relevant for quantum applications. Considering the importance of MCMC, showing how to obtain quadratic speedups using quantum-accelerated MCMC for nonlinear problems is valuable and may be useful more broadly in areas outside of the quantum algorithms community.

**Limitations:**

Yes.

**Paper Formatting Concerns:**

No major formatting concerns.

**Quality:**

3

**Strengths And Weaknesses:**

## Strengths
- The problem of non-linear nested expectations is an interesting one with several real-world applications. The question of whether quantum computers can provide a quadratic speedup is a very natural and non-trivial one, requiring some technical innovations.
- The discussion of why quadratic speedup is lost when naively applying quantum-accelerated Monte Carlo is very insightful.
- The idea of quantum-inside-quantum appears to be a useful idea that may have applications elsewhere.

## Weaknesses
- (Minor typo?) On page 2, it looks like $\gamma$ and $\psi$ are both being used to represent the conditonal expectation of $\phi(x,Y)$.
- As it stands, Figure 1 is a bit difficult to interpret and could be improved. I suppose the idea here is that QA-MC is being "distributed" more evenly in Q-NestExpect compared to QA-MLMC, but this is somewhat vague and hard to understand.
- While it is not a major issue, I think this paper could benefit from some numerics to validate the theoretical findings.
- In the context of quantum algorithms, a quadratic speedup is theoretically interesting but practically disappointing since the overhead of error correction is expected to outweigh any possible quantum advantage.

---

> ### Author Rebuttal · Authors · 2025-07-28
>
> We thank the reviewer for the positive feedback! We are encouraged that the reviewer thinks the problem is interesting and the discussion is insightful. We address the questions below:
>
>
> **Better than quadratic speedup**: Great question. We are also very interested in exploring this further. Currently, we don’t have concrete examples or applications. The main challenge is that all the subroutines so far rely on Grover’s search, which offers at most a quadratic speedup. However, in these multilevel problems, the allocation of computational resources also impacts the total cost, so there may still be some potential that a superquadratic speedup comes from (quadratic speedup on all the subroutines + an overall more efficient allocation strategy).
>
> A side note is, there are many Monte Carlo problems that have worse than $O(\epsilon^{-2})$ cost. The simplest one is to estimate the expectation of a heavy-tailed distribution with finite $(1+\delta)$-th moment but not 2nd moment. However, in that case, the quantum algorithm can still only provide a quadratic speedup (their corresponding cost will also be slower than the typical $O(\epsilon^{-1})$ [1].
>
>
> **One single loop?**: Good point. Currently, we believe the Q-NESTEXPECT algorithm cannot be reinterpreted as a single quantum-accelerated Monte Carlo algorithm without sacrificing its performance advantage. The Q-NESTEXPECT algorithm is built on improving the classical MLMC method, which itself isn’t a single loop (the core idea of MLMC is to break the target quantity into components and estimate each piece separately). This splitting strategy is crucial to the performance advantage of MLMC compared to simpler nested Monte Carlo methods [2].
> Thus, if a classical design were to achieve the same performance as MLMC using a single loop, there might be a possibility to design a quantum algorithm (or reinterpret our existing algorithm) as a single quantum-accelerated Monte Carlo algorithm. However, this is not the case as of now.
>
>
>
>
>
> **Typo**: Thanks for catching this! We will fix in the revised version.
>
> **Better Figure 1**:  Thank you for the comment. As suggested (and also suggested by reviewer ZtmP), we will revise Figure 1 in the updated version to ensure it is clearer and more informative. We will also include additional explanations to help readers interpret the figure more easily.
>
>
>
> [1] Quadratic Speed-up in Infinite Variance Quantum Monte Carlo, Blanchet et. al. 2024.
>
> [2] On nesting Monte Carlo estimators, Rainforth et. al., ICML 2018.

---

> > ### Comment · Reviewer_H1SM · 2025-08-04
> >
> > I thank the authors for the rebuttal, which adequately addresses my questions and concerns. I will raise my score to 5.

---

### Official Review · Reviewer_ZtmP · 2025-07-03

**Clarity:** 4
**Significance:** 3
**Originality:** 3
**Rating:** 5
**Confidence:** 4

**Summary:**

The main focus of this submission is the estimation of non-linear functions of probability distributions. More specifically, the authors proposed quantum algorithms based on an MLMC framework to estimate nested expectations. This proposed quantum algorithm improved upon the previous best-known quantum algorithm (Due to An et al in 2021), and it achieves optimal complexity. The core of the algorithm (and the source of the quantum speedup) is from the quantum mean estimation algorithm by Kothari and O'Donnell in 2023.

**Questions:**

1. In Section 1.1, the authors listed four potential applications, but in Section 4, only three of them were discussed. What about the conditional stochastic optimization?

2. The algorithm assumes constants $V$ and $S$ are known. In applications such as Bayesian experiment design, EVPPI, and CoC option pricing, how are these constants readily known?

3. Line 190: Maybe the authors should provide some intuition about the garbage register. I know what they are, but people not familiar with quantum computing might be confused.

Minor comment: the quality of Figure 1 is pretty low. Maybe the authors should consider using a better image.

**Ethical Concerns:**

["NO or VERY MINOR ethics concerns only"]

**Final Justification:**

Final score: 5

My questions have been addressed in the rebuttal. I keep my positive score.

**Limitations:**

yes

**Quality:**

4

**Strengths And Weaknesses:**

This submission is very well-written. I personally appreciate Section 3.1, where a high-level overview of the main contributing factors of the complexity, as well as the intuition of the MLMC structure. I believe this is beneficial to the readers not familiar with quantum MCMC-type algorithms.

In terms of results, the main strength of this submission is the newly designed MLMC sequence, which enables the use of quantum mean estimation to estimate the expectation of each level (outside quantum), which further uses quantum mean estimation (inside quantum). For applications with nested expectations, it is quite natural to use quantum mean estimation inside quantum mean estimation.

---

> ### Author Rebuttal · Authors · 2025-07-28
>
> We thank the reviewer for the positive feedback! We are encouraged that the reviewer thinks our paper is well-written and clear.
>
> We address the questions raised below:
>
> **Conditional stochastic optimization**: Good point. Conditional stochastic optimization (CSO) focuses on solving $\max_x F(x)$, where each fixed $x$ involves a nested expectation. We did not include this in Section 4 of the submitted version, as it introduces an additional layer of complexity (optimization) beyond straightforward nested expectations in other applications and was also excluded due to space limitations.
>
> Following the suggestion, we will provide a rigorous formulation of the CSO in the revise version. We will include details on the space of $x$ and assumptions on $F$, and explain how our results lead to improvements over classical results.
>
> **Constant in these applications**: Thanks for the question. We acknowledge that the assumption on a known upper bound for the second moment may not always hold, or difficult to verify in many cases. In practice, it usually can be verified when the function is bounded (for instance, when $g$ represents a risk or payoff that is inherently bounded) or when the random variables $(X,Y)$ live in a  compact space (e.g., $[0,1]^2$).
>
> An interesting direction for future research is to relax the assumption of known constants to simply requiring finiteness (i.e., $< \infty$; see also our response to reviewer hqsW). In particular, we note that the assumption of finite variance with a known constant is also common in other quantum mean estimation applications, such as stochastic optimization [5]. Therefore, relaxing these assumptions could be beneficial, potentially strengthening the results in these contexts as well.
>
> **Explain garbage register**: Thanks! We will do this in the revised version.
>
> **Better Figure 1**: Thank you for the comment! We will revise Figure 1 in the updated version to ensure it is clearer and more informative.
>
> [1] Quantum speedups for stochastic optimization, Sidford and Zhang, NeurIPS 2023.

---

> > ### Comment · Reviewer_ZtmP · 2025-08-06
> >
> > Thanks for addressing my questions. I hope the authors will incorporate the promised changes in the final version.

---

### Official Review · Reviewer_hqsW · 2025-07-03

**Clarity:** 4
**Significance:** 3
**Originality:** 3
**Rating:** 5
**Confidence:** 4

**Summary:**

This paper proposes a novel quantum algorithm called "quantum-inside-quantum MLMC" that achieves a quadratic speedup for estimating nested expectations—a class of non-linear functionals arising in various practical scenarios such as Bayesian experiment design and financial engineering. Classical Monte Carlo methods typically have a complexity of $O(\epsilon^{-2})$, and existing quantum methods reduce this to $O(\epsilon^{-1.5})$. The proposed approach further improves it to a nearly optimal $O(\epsilon^{-1})$, leveraging specifically designed multilevel Monte Carlo decomposition optimized for quantum computing.

**Questions:**

- Can the authors comment on how the algorithm’s complexity would degrade or change if the assumptions (such as Lipschitz continuity) were weakened or violated?
- For the application of MLMC in finance and optimization listed in Section 5, does the proposed quantum algorithm potentially preserve the quadratic speedup? Especially consider the assumed conditions in page 5.
- The paper focuses primarily on nested expectations. Could the authors elaborate on whether and how their quantum-inside-quantum MLMC approach might extend to more general classes of nonlinear functional estimation problems?

**Ethical Concerns:**

["NO or VERY MINOR ethics concerns only"]

**Final Justification:**

I think this paper is theoretically solid and has various applications in finance and optimization. The techniques applied in this paper are  of independent interest. Also the authors address my concerns on the assumptions during the rebuttal period. I would recommend for acceptance.

**Limitations:**

Yes.

**Paper Formatting Concerns:**

No.

**Quality:**

3

**Strengths And Weaknesses:**

#### Strengths

- The authors propose a novel quantum algorithm that specifically leverages quantum advantages at multiple hierarchical levels of the Monte Carlo estimation procedure, rather than merely accelerating classical MLMC subroutines individually.
- The authors identify and address the limitation of direct quantum adaptations of classical non-linear MLMC, which clearly explain why the quadratic speedup is lost and how they recover the speedup.
- Achieving a near-optimal complexity of $\tilde{O}(\epsilon^{-1})$, confirmed with known lower bounds, represents significant theoretical progress and firmly establishes quantum advantage in this domain.

#### Weaknesses

- The complexity analysis and algorithmic performance depend heavily on the assumptions listed in page 5. In practical scenarios, these conditions may not strictly hold or may degrade, which may limits the applicability of the proposed algorithm.
- While theoretically impressive, the implementation nested quantum subroutines is potentially complex. Details on practical quantum resources (circuit depth and width, error accumulation, noise resilience, quantum memory requirements, etc.) required to realize the quantum-inside-quantum approach on real or near-term quantum hardware remain largely unexplored.

---

> ### Author Rebuttal · Authors · 2025-07-28
>
> We thank the reviewer for the valuable suggestions! We address the questions raised below:
>
> **Assumptions and Practicality**:
>
> Good point. While the assumptions we made appear frequently in the literature (e.g., the original nested expectation paper [1]), we acknowledge that they may not hold perfectly in all practical scenarios. There are essentially two types of technical assumptions appeared in the current version:
> 1) *Smoothness assumption (Assumption 1);*
> 2) *Moment assumption (Assumption 2,3).*
>
> We discuss them separately.
>
> *Regarding the smoothness assumption*: It is indeed possible to generalize this assumption. Additional knowledge about the functions or the underlying distribution could further relax the continuity requirement. In classical Monte Carlo literature, there are studies on nested expectations where the function $g$ in our notation is a Heaviside function [3,4]. The idea of their assumptions is to assume that the underlying distribution behaves appropriately at the discontinuous points, ensuring that it does not cause issues.
> In summary, there is a tradeoff between the assumptions on distributions and the regularity assumptions on the functions. Stronger distributional assumptions and/or extra knowledge on the function can lead to weaker assumptions on its smoothness, and vice versa. If we use the same distribution assumptions as in [3,4] and assume $g$ is a Heaviside function, we could still get the quadratic speedup.
>
> *On the necessity of finite second moments:* A finite second moment may be required to achieve a complexity of $\tilde{O}(\epsilon^{-1})$. Reference [2] shows, even in the context of standard Monte Carlo estimation, assuming only the existence of a $(1 + \delta)$-th moment with $\delta \in (0,1)$ is insufficient for attaining $\tilde{O}(\epsilon^{-1})$ cost with a quantum algorithm (but a quadratic speedup still exists).
>
> *On the known constants for the second moment/variance:* There exists an interesting intermediate regime where the assumption (1-2) could be replaced by $<\infty$ (instead of the conditions $\leq V$ and $\leq S$, where $V$ and $S$ are constants known to the users in advance.) Right now we do not know if we can design a quantum algorithm to achieve $\tilde O(\epsilon^{-1})$ cost in this relaxed assumption.
> One potential approach is to first perform a low-cost Monte Carlo estimation to obtain an estimated upper bound that holds with high probability (e.g., via the Markov inequality), and then plug-in this estimated bound into our algorithm.
> A rigorous analysis of these costs is likely complex, but it is certainly a worthwhile direction for future work. In particular, we note that the assumption of finite variance with a known constant is also common in other quantum mean estimation applications, such as stochastic optimization [5]. Therefore, relaxing these assumptions could be beneficial, potentially strengthening the results in these contexts as well.
>
>
>
> **Validity of the assumptions on finance and optimization applications:**
> The smoothness assumption is commonly satisfied in finance (particularly in option pricing) and optimization problems. In such contexts, the function $g$ is typically smooth or piecewise linear [6]. It is not always the case, though, especially when $g$ could be a Heaviside function [3, 4].
>
> The finiteness of the second moment with a known constant also holds in many applications, as seen in [5, 7, 8]. In practice, it usually can be verified when the function is bounded (for instance, when $g$ represents a risk or payoff that is inherently bounded) or when the random variables $(X,Y)$ live in a  compact space (e.g., $[0,1]^2$). However, we acknowledge that these assumptions may not always hold, or at least may be difficult to verify outside of these scenarios.
>
> **Other applications on non-linear functionals**:
>
> Thank you for the question. The main idea is that the problem should have a compositional structure (such as the inside and outside expectations in our case), which allows us to leverage an effective resource allocation strategy.
>
> There are potential generalizations relevant to optimization that we have in mind: the first is the Conditional Stochastic Optimization (CSO) problem [7], where the goal is to optimize $\max_x F(x)$, and each $F(x)$ corresponds to a nested expectation. Our algorithm can be directly applied to estimate $\hat{F}(x)$ for $F(x)$. This estimate can then be combined with gradient descent (if $F$ is strongly convex and $x$ belongs to a continuous space) or with quantum maximum finding (if $x$ is from a discrete set). Similarly, our method might also be useful for Stochastic Composition Optimization (SCO) [9].
>
> Thanks again for your valuable feedback!
>
>
> [1] On nesting Monte Carlo estimators, Rainforth et. al., ICML 2018.
>
> [2] Optimal randomized multilevel Monte Carlo for repeatedly nested expectations, Syed and Wang, ICML 2023.
>
> [3] Multilevel Nested Simulation for Efficient Risk Estimation, Giles and Haji-Ali, SIAM Journal on Uncertainty Quantification 2019.
>
> [4] Nested Simulation in Portfolio Risk Measurement, Gordy and Juneja, Management Science, 2010.
>
> [5] Quantum speedups for stochastic optimization, Sidford and Zhang, NeurIPS 2023.
>
> [6] Multilevel Simulation of Functionals of Bernoulli Random Variables with Application to Basket Credit Derivatives, Bujok et. al., Methodology and Computing in Applied Probability, 2013.
>
> [7] Biased Stochastic First-Order Methods for Conditional Stochastic Optimization and Applications in Meta Learning, Hu et. al., NeurIPS 2020.
>
> [8] Bias-Variance-Cost Trade-off of Stochastic Optimization, Hu et. al., NeurIPS 2021.
>
> [9] Accelerating stochastic composition optimization, Wang et. al., JMLR 2017.

---

> ### Comment · Reviewer_hqsW · 2025-08-05
>
> I thank the authors for the detailed response that addresses my concerns. I will raise the score to 5.

---

### Comment · Area_Chair_vSGu · 2025-08-04
**Author-Reviewer Discussion Period Ending Soon**

Dear Reviewers,

The Author-Reviewer discussion period has begun and will end on **August 6**. Please read the rebuttal and let us know whether it satisfactorily addresses your concerns. If not, could you specify what remains inadequate? Your response will help us evaluate the paper and assist the authors in improving their work.

Please avoid responding at the last minute, as the authors may not have sufficient time to clarify your concerns.

Thank you!

Best,
AC

---

### Decision · Program_Chairs · 2025-09-17

**Decision:**

Accept (spotlight)

**Comment:**

Consider the problem of estimating a nested expectation. This paper proposes a quantum algorithm with cost $\tilde{O}(\epsilon^{-1})$. Previously, the best-known classical MLMC algorithm has a cost of $\tilde{O}(\epsilon^{-2})$, and the best-known quantum algorithm, QA-MLMC (An et al., 2021), has a cost of $\tilde{O}(\epsilon^{-1.5})$.

The paper is very well written, with an inspiring discussion of the ideas in previous and current works. The problem considered is adequately important, and the improvement over existing work is significant. Reviewer BADi correctly pointed out that the application of the proposed algorithm is more restrictive than that of QA-MLMC by An et al. (2021). Nevertheless, I believe this paper is already worthy of publication given its contribution to the specific problem of nested expectation estimation. The other reviewers also appear to be positive about this paper.

Given the above, I suggest accepting this paper as a spotlight.